# Towards a Better Understanding of Genotype–Phenotype Correlations and Therapeutic Targets for Cardiocutaneous Genes: The Importance of Functional Studies above Prediction

**DOI:** 10.3390/ijms231810765

**Published:** 2022-09-15

**Authors:** Mathilde C. S. C. Vermeer, Daniela Andrei, Luisa Marsili, J. Peter van Tintelen, Herman H. W. Silljé, Maarten P. van den Berg, Peter van der Meer, Maria C. Bolling

**Affiliations:** 1Department of Cardiology, University Medical Center Groningen, University of Groningen, 9713 AV Groningen, The Netherlands; 2Department of Dermatology, University Medical Center Groningen, University of Groningen, 9713 AV Groningen, The Netherlands; 3Clinique de Génétique, Hôpital Jeanne de Flandre, CHU de Lille, 59000 Lille, France; 4Department of Genetics, University Medical Center Utrecht, Utrecht University, 3508 AB Utrecht, The Netherlands

**Keywords:** cardiocutaneous syndromes, genotype–phenotype correlation, functional analysis of genetic variants

## Abstract

Genetic variants in gene-encoding proteins involved in cell–cell connecting structures, such as desmosomes and gap junctions, may cause a skin and/or cardiac phenotype, of which the combination is called cardiocutaneous syndrome. The cardiac phenotype is characterized by cardiomyopathy and/or arrhythmias, while the skin particularly displays phenotypes such as keratoderma, hair abnormalities and skin fragility. The reported variants associated with cardiocutaneous syndrome, in genes *DSP, JUP, DSC2, KLHL24, GJA1*, are classified by interpretation guidelines from the American College of Medical Genetics and Genomics. The genotype–phenotype correlation, however, remains poorly understood. By providing an overview of variants that are assessed for a functional protein pathology, we show that this number (n = 115) is low compared to the number of variants that are assessed by in silico algorithms (>5000). As expected, there is a mismatch between the prediction of variant pathogenicity and the prediction of the functional effect compared to the real functional evidence. Aiding to improve genotype–phenotype correlations, we separate variants into ‘protein reducing’ or ‘altered protein’ variants and provide general conclusions about the skin and heart phenotype involved. We conclude by stipulating that adequate prognoses can only be given, and targeted therapies can only be designed, upon full knowledge of the protein pathology through functional investigation.

## 1. Introduction

Pathogenic variants in genes encoding for proteins expressed in both skin and heart may cause so-called cardiocutaneous syndromes. These can be inherited in an autosomal dominant or recessive way, and can be life-threatening [1,2,3]. Cardiocutaneous syndromes usually present in childhood with the solitary or combined skin and adnexal features of skin fragility, palmoplantar keratoderma (PPK), woolly hair (WH) and alopecia [3]. When present, the skin fragility phenotype usually persists throughout life and presents as blisters, erosions and wounds with an intraepidermal level of skin separation [2,4,5,6,7]. Cardiomyopathy usually manifests later in adult life, with a high risk of death through arrhythmias or end-stage heart failure [8]. While arrhythmogenic cardiomyopathy (ACM) is the most commonly observed form, with a prevalence of 1:5000, other cardiomyopathies such as dilated (DCM; 1:250), hypertrophic (HCM; 1:500) or non-compaction (NCCM) have also been observed in patients. The underlying cause of this disease is mainly found in genes that encode for commonly shared proteins in the skin and heart, involved in the intercellular compliance network. This network consists of intracellular intermediate filaments anchored to specialized cell–cell connecting structures, called desmosomes. In 2000, the first desmosomal gene variant underlying a cardiocutaneous syndrome was identified as a homozygous *DSP* variant that translated to a C-terminal truncated desmoplakin protein [1]. Today, many variants have been associated with a disease phenotype, found in desmosomal genes *DSP* (also known as Carvajal syndrome [9,10]), *JUP* (also known as Naxos disease [9,11]) and *DSC2* [2], the intermediate filament regulating gene *KLHL24* [12], and the gap junction gene *GJA1* [13] (Figure 1).

While most variants in the abovementioned genes cause a dual organ phenotype, some variants are strictly confined to a skin or cardiac phenotype. This phenomenon is poorly understood. Guidelines to interpret pathogenicity of genetic variants, provided by the American College of Medical Genetics and Genomics (ACMG) [14], are currently multifaceted, meaning they take population-based frequencies, segregation analysis, prediction models and functional evidence into consideration. While population-based frequencies and segregation analysis determine disease penetrance, they do not unveil the underlying mechanism. In reality, the pathogenicity of variants is predicted by and dominated through in silico algorithms that rely on physical properties of amino acids, the conservation of residues in sequence alignments of closely related proteins or sequence and structural comparisons. However, in silico algorithms frequently contradict one another, and the actual impact on protein production and function can only be determined via functional assays. As the genotype–phenotype correlations of cardiocutaneous genes are poorly understood and therefore prognoses are difficult to establish, there is a pressing unmet need for fundamental research that determines the effect of genetic variants on the protein (function), which in turn is also paramount to the development of targeted therapies [15].

In this review, we summarized the functional evidence of 115 investigated variants in cardiocutaneous-expressed genes and compared these with their ACMG class and the prediction on protein pathology through in silico algorithms. Based on this overview and the major findings, we provide recommendations for therapeutic strategies and future directions.

## 2. Desmosomal Genes

Desmosomes are mirroring, transmembrane protein chains that connect the intermediate filament networks of neighbouring cells. Each chain continuously (dis)assembles due to the turnover of five desmosomal protein types: desmoplakin, plakoglobin, plakophilins, desmocollins and desmogleins (Figure 1) [16]. The expression of two genes is critical to the formation of all desmosomes: namely *DSP,* encoding two differently spliced desmoplakin proteins (DPI and DPII) and *JUP,* encoding plakoglobin (PG). Meanwhile, plakophilins, desmocollins and desmogleins are expressed in a tissue-specific manner and are therefore encoded by multiple genes. The plakophilin and desmocollin gene family each contain three subtypes (*PKP1, PKP2* and *PKP3,* encoding proteins PP1, PP2 and PP3 and *DSC1, DSC2* and *DSC3,* encoding proteins DC1, DC2 and DC3), while the desmoglein gene family contains four subtypes (*DSG1, DSG2, DSG3* and *DSG4,* encoding proteins DG1, DG2, DG3 and DG4). Keratinocytes of the skin and adnexes express both DP isoforms and PG, in addition to any of the aforementioned PP1-3, DC1-3 and DG1-4 combinations. Each specific composition is in accordance with the differentiation status of keratinocytes in the epidermis [17]. Desmosomal proteins are crucial for epidermal integrity and proper epidermal proliferation and differentiation, and irregularities thereof may cause a skin phenotype [18]. Desmosomal proteins also accommodate hair growth and irregularities thereof. The hair follicle contains keratinocytes in an inner and an outer root sheath. In straight hairs, shafts are straight and homogenous, without clear delimitations, but in coiled hair, these shafts have retro-curvatures [19,20,21]. This curve is achieved via a rotation mechanism of aberrantly proliferating and differentiating cells in the inner root sheath [18,22]. Unlike the skin, the protein composition of desmosomes in cardiac tissue is fixed and consists of DPI, PG, PP2, DC2 and DG2. In the heart, desmosomal anomalies typically disrupt the mechanical continuity of cardiac muscle fibres, which is essential for proper conductance and cardiac muscle contraction. Desmosomes are crucial for the anchorage of cardiomyocytes at the intercalated disc parallel to the direction of strain, where they internally dock the desmin network [16]. Thus far, variants in desmosomal genes *DSP*, *JUP* and *DSC2* have been associated with a cardiocutaneous phenotype.

### 2.1. Reported DSP Variants

*DSP* encodes for two differently spliced DP proteins: DPI (332 kDa) and a smaller DPII isoform (260 kDa) that contains a shorter rod domain [23]. The latter is created by an alternative donor splice site in exon 23. The N-terminal plakin-domain binds with PPs and PG, while domains B and C at the C-terminal side bind to intermediate filaments (Figure 2). In cardiac muscle, *DSP* is predominantly spliced into DPI, while the skin contains both isoforms equally. The ClinVar database reported 3290 variants in *DSP*. Of these, 495 were claimed (likely) pathogenic; 1026 (likely) benign; 161 show conflicting interpretations and 1608 have an unknown significance. Only 48 variants were substantiated by functional evidence, including data from transgenic mouse models (see Figure 2; full report in Table 1). This indicates that over 98% of all *DSP* variants were merely predicted by in silico algorithms. For the majority of variants (36/48), the predictions on protein level were correct, while only partially correct in 2/48 variants and incorrect in 5/48 variants. In 5/48 variants, the functional evidence was too elusive to draw conclusions. Moreover, the in silico prediction algorithms frequently contradicted one another, providing little help in assessing the pathogenicity of *DSP* variants.

#### 2.1.1. *DSP* Variants Causing DP Reduction

Complete loss of DPI&II is probably incompatible with human life, as it causes early embryonic lethality in mice [55]. More importantly, it has not been functionally observed in human patients. However, several variants that cause protein reduction (<100% native DPI/DPII left) in humans and animal models have been reported [56,57,58]. In total, 9/48 variants resulted in variable degrees of DP reduction (Table 1) and inflicted disease in either a dominant (n = 5) or recessive (n = 4) mode of inheritance. This occurred due to two splice-site variants (c.273 + 5G > A and c.939 + 1G > A), four nonsense-inducing variants (c.699G > A, c.3799C > T, c.3805C > T and c.5208_5209del) located before the terminal exon, and one nonsense-inducing variant in the terminal exon (c.6687del) that was unexpectedly targeted by NMD. Moreover, one missense (c.1348C > G) and one in-frame indel variant (c.969_974del) resulted in DP protein reduction, probably due to instable protein degradation. All of the aforementioned variants affected both isoforms, except for variants c.3799C > T, c.3805C > T and c.5208_5209del. The latter are located in the ROD domain of DPI and therefore only affect DPI, but not DPII. Interpreting the phenotype of patients, DP deficiency (≤50% native DPI) seems to be associated with severe cardiomyopathy, while DP deficiency in the skin (≤50% native DPI and DPII) is mostly associated with PPK and WH. Recessive variants that caused loss of DPI but not DPII, or extreme deficient levels of both DPI and DPII (<20%), were associated with skin fragility [41].

#### 2.1.2. *DSP* Variants Causing an Altered DP Protein

The majority of functionally investigated *DSP* variants (36/48) led to an altered DP protein (Table 1), due to 23 dominantly and 13 recessively inherited variants. Missense variants were the predominate source for altered DP proteins (n = 27), while the remaining nine variants were due to nonsense or nonsense-inducing variants. Out of the 36 variants, two variants were located near the N-terminus, ten were located in the plakin domain, three in the ROD domain of DPI, six in domain A, five in domain B, two in the linkers, five in domain C and three near the C-terminus. All but one (c.6687del) of the nonsense-inducing variants in the terminal exon of *DSP* skipped NMD. As expected, the phenotype of patients with an altered DP protein indicated that a recessive mode of inheritance was more severe than a dominant mode of inheritance, mostly due to absence of native protein in the former. Cardiomyopathy was observed in 31/36 of the variant carriers, while it went unobserved or unreported in the others. In the skin, 11/36 variants resulted in PPK (recessive n = 9, dominant n = 2) often with WH (recessive n = 7, dominant n = 2). However, PPK and WH were frequently not observed or not reported in the studies primarily focused on the cardiac phenotype. Furthermore, 10/36 variants caused skin fragility, mostly due to a recessive inheritance (n = 7, dominant n = 3). Variants located in the plakin domain frequently affected the binding efficiency to PG [24,25,26], PPs [59] or intermediate filament anchorage [31,60]. Variants located in domains A, B or C almost always affected the binding affinity to intermediate filaments, especially when causing severe property alterations in domains B and C or loss of these through C-terminal truncation. The functional evidence of the remaining 3/48 variants was inconclusive as to whether it resulted in protein reduction or an altered DP protein.

#### 2.1.3. Potential Therapeutic Avenues

Given the contradicting results from in silico algorithms and several inconsistencies between the prediction and functional evidence, functional assays of the remaining *DSP* variants would be strongly encouraged. The nine variants causing DP reduction indicated that the disease severity tends to be dose-dependent in nature, both in the heart as well as in the skin. Hence, strategies to increase native DP protein levels, especially in the heart, would be of benefit to patients. Injections with *DSP* mRNA in DP-deficient zebrafish have been promising in regaining cardiac function [61]. Besides strategies like RNAi or CRISPR that eliminate protein expression from mutant alleles in patients with altered DP proteins, strategies should simultaneously aim to increase native DP protein levels. For most of the functionally investigated variants, it remains unclear whether they cause a dual organ phenotype, as it is not often assessed or specified.

### 2.2. Reported JUP Variants

The *JUP* gene encodes for the 82 kDa PG protein, also known as ɣ-catenin. PG contains an N-terminal head-domain, 12 armadillo domains and a C-terminal tail-domain (Figure 3). PG belongs to the catenin protein family and is highly homologous to β-catenin, a potent transcription factor of the canonical Wnt/β-catenin signalling pathway. PG is an important desmosomal protein, comprising the outer dense plaque of the desmosome and connecting the transmembrane DG and DC proteins to DP and PP. β-catenin and PG can be substituted for one another, as β-catenin can be incorporated into desmosomes, while PG can also act as a nuclear transcription factor [62]. The ClinVar database has reported 838 variants in *JUP*. Of these, 30 are claimed (likely) pathogenic; 307 (likely) benign; 70 show conflicting interpretations, and 431 have an unknown significance. Merely eight variants where substantiated by functional evidence, including data from transgenic mouse and zebrafish models (see Figure 3 and full report in Table 2). As for *DSP* variants, effects of over 98% of all *JUP* variants were merely predicted by algorithms. The predictions on protein level were correct in 4/8 variants, incorrect in 3/8 variants, while the functional data remained inconclusive in 1/8 variants.

#### 2.2.1. *JUP* Variants Causing PG Reduction

Complete loss of PG induces lethality within embryogenesis in mice due to severe heart defects or immediately post-natal due to severe skin fragility [74,75]. Highly suppressed PG protein levels (<10%) also lead to ACM [76], while 40% protein levels do not induce cardiac dysfunction in mice [77]. This suggests that threshold levels for cardiomyopathy in mice span between 10–40% of native protein. Two human variants (c.201del and c.1615C > T, nonsense) were reported to induce complete PG depletion in homozygous patients [66,68]. These carriers developed severe and sometimes lethal skin fragility, in combination with PPK and alopecia in homozygous c.201del carriers. Cardiomyopathy was not observed in any but one patient with old age [66]. Meanwhile, no other patients with PG reduction (≤50% protein) have been reported. The above results suggest that (near) PG depletion is strongly correlated to skin fragility. However, the dose effect of PG reduction on the development of skin features is unknown and warrants further functional studies. In contrast, while PG depletion induces cardiac lethality during embryogenesis in mice, the limited data currently suggest that the human heart may be protected, perhaps due to functional compensation of β-catenin [62,71]. To accurately assess the cardiac penetrance in patients, more variants predicted to cause reduced or depleted PG levels need to be investigated.

#### 2.2.2. *JUP* Variants Causing an Altered PG Protein

Five functionally investigated variants (5/8) resulted in an altered PG protein (Table 2). These resulted from three recessively inherited nonsense-inducing variants, resulting in either a N-terminal (Glu2_Met43del) or C-terminal (Trp680Glyfs*11 and Met686Asnfs*5) truncated protein. In addition, two dominantly inherited variants, Ser39dup and Arg577Cys, caused ACM and resulted in proteins comparable to the size of native PG. Unlike the C-terminal truncations, the recessive N-terminal truncation induced skin fragility with PPK and WH, but no cardiac dysfunction. The variant causative for this N-terminal truncation c.71C > A, introduces a PTC at Ser24, but translation re-initiation took place at position Met43, which resulted in deletion of the first 42 amino acids. This suggests that any nonsense-inducing variant located between *JUP*:c.1_126 will likely cause translation re-initiation and a similar phenotype and effect on protein. Opposingly, the two recessive C-terminal truncations correlated with ACM, PPK and WH, but did not induce skin fragility in patients. Moreover, two homozygous PG mouse knockin Trp680Glyfs*11 models were developed, one with and one without fusion of the final five exons. In mice without fusion of exons, this variant resulted in NMD, and only very low levels of C-terminal truncated protein were expressed. These mice died on postnatal day one due to severe skin fragility, induced by depleted PG. Due to their short lifespan, the effect on cardiac function in later stages of development is unclear. In mice with fusion of exons, high levels of C-terminal truncated protein were observed, similarly as in patients. Nonetheless, even at 11 months of age, mice failed to develop cardiac dysfunction [71]. These data suggest that there may be little resemblance between the cardiac function of humans and mice with regard to *JUP* variants. More variants need to be functionally investigated to draw definitive conclusions.

#### 2.2.3. Potential Therapeutic Avenues

Currently, too little functional evidence is available to adequately address potential therapeutic strategies that would benefit the cardiac function of patients with disease-causing *JUP* variants. Meanwhile, strategies to increase native protein in patients with skin fragility seem appropriate for all patients with PG depletion and patients with PG proteins that lack the N-terminus. Whether a skin phenotype is only observed in the case of biallelic *JUP* variants, as the eight functionally investigated variants now suggest, needs additional functional evidence. Furthermore, almost half of the functionally investigated variants were falsely predicted, which further pressed the need for more functional studies.

### 2.3. Reported DSC2 Variants

The *DSC2* gene encodes for two transmembrane cadherin isoforms: DC2a (99 kDa) and DC2b (93 kDa). The DC2a isoform contains the complete intracellular segment (ICS), whereas this domain is 53 amino acids shorter in DC2b [78], due to alternative splicing of exon 16 (Figure 4). Both isoforms are first processed into a precursor protein, followed by a mature protein that can be incorporated into desmosomes. Maturely processed DC2 serves as a transmembrane desmosomal protein, important for extracellular cell–cell attachment. The ClinVar database has reported 1209 variants in *DSC2*. Of these, 102 were claimed (likely) pathogenic; 409 (likely) benign; 85 show conflicting interpretations and 613 have an unknown significance. Notably, merely 15 variants were substantiated by functional evidence, including data from transgenic mouse and zebrafish models (see Figure 4 and full report in Table 3). The same trend seen for *DSP* and *JUP* variants, is also observed for *DSC2*, indicating that over 98% of all variants have not been functionally investigated. The predictions on protein level were correct in 11/15 variants, while incorrect in 3/15 variants and unclear in 1/15 variants.

#### 2.3.1. *DSC2* Variants Causing DC2 Reduction

No patients with complete absence of DC2 protein have been reported. Instead, four variants (4/15), located before the terminal exon, resulted in both DC2a and DC2b protein reduction. The recessively (c.1913_1916delAGAA; ≤10% protein left [87]) and dominantly (c.631-2A > G; 40% protein left) inherited nonsense-inducing variants both caused ACM in patients. Compound heterozygosity of out-of-frame indel variant c.140_147delAACTTGT resulted in NCCM and hypertrophy, which is the only functionally investigated *DSC2* variant associated with cardiomyopathy other than ACM [80]. The missense variant c.394C > T caused 50% protein reduction via instable protein degradation and caused ACM in a dominant mode of inheritance. Altered electrical properties, a key characteristic of ACM, have been observed in patient hiPSC-CMs containing this missense variant [82]. Moreover, dominantly inherited variants c.394C > T and c.631-2A > G were also investigated in a zebrafish model. The ACM phenotype in both models was rescued by injecting human wildtype but not mutant *DSC2* mRNA [82,85]. One of these studies additionally showed that gradual knockdown of *DSC2* resulted in dose-dependent cardiac disease severity [85]. This seems to corroborate with the human data, suggesting that cardiomyopathy occurs in situations with ≤50% of native DC2 protein: and the higher the reduction, the more severe the phenotype. Furthermore, in mice, neither complete nor heart-specific knockout of *DSC2* resulted in any altered viability or cardiac phenotype [92], which emphasizes differences in disease susceptibility among species. None of the DC2 protein-reducing variants caused a skin phenotype, indicating that near loss of the DC2 protein is well tolerated by the skin. Based on the few investigated variants and contradicting results of animal models, incisive conclusions are still difficult to draw.

#### 2.3.2. *DSC2* Variants Causing an Altered DC2 Protein

Ten variants (10/15) resulted in an altered DC2 protein (Table 3), which predominantly caused ACM via a dominant (n = 7) or recessive (n = 3) mode of inheritance. Heterozygous variant c.-1445G > C in the 5′UTR affected transcription factor binding mechanisms. Meanwhile, five variants had pronounced effects on the processing of DC2 precursor proteins. For instance, artificial transfection experiments containing missense variants Glu102Lys, Arg203Cys, Thr275Met and Ile345Thr showed punctate cytoplasmic staining, with no or partial ability to be incorporated into desmosomes [84,86,90]. Moreover, nonsense variant Gln554* escaped NMD and resulted in a C-terminal truncated protein, affecting both isoforms [86]. This variant also affected the processing of DC2 precursor proteins, and while a small proportion of maturely processed proteins was incorporated into the desmosome, a larger proportion of precursor proteins remained in the cytoplasm. It is still uncertain whether missense variant Ile520Thr will induce similar alterations that affect the processing of DC2 precursor proteins [80]. Opposingly, nonsense-inducing variants, Asp852Thrfs*4 and Ala897Lysfs*4, only affected isoform DC2a and caused a C-terminal truncated protein. Both were fully processed into a mature protein form, were incorporated into desmosomes, but lost their ability to bind to DP [84] and PG [84,90]. The latter suggests a similar perturbing binding interface for Gln554*. No protein processing information is available on variant Gly790del, other than that it is expressed and translated into a transgenic mouse model. Neither the heterozygous nor homozygous mice showed structural or functional defects in the ventricles or lethal arrhythmias, and only homozygous aged mice showed slight left ventricular dysfunction. This mouse model therefore does not represent the phenotypic severity of the heterozygous Gly790del patients with ACM. In most (7/10) variants, apart from recessive variants Gln554*, Thr275Met and Ile520Thr, ACM was observed in a dominant mode of inheritance. Only recessive inheritance of variant Ser614Ilefs*11 caused PPK and WH, but the functional data were unclear as to whether it causes protein reduction or an altered protein function in patients [87,88]. A skin phenotype was furthermore not observed or went unreported in the other variants.

#### 2.3.3. Potential Therapeutic Avenues

Whether *DSC2* variants can truly cause a cardiocutaneous phenotype remains somewhat elusive, given that only one investigated variant was associated with PPK and WH, and others that do associate with a skin phenotype were not investigated [88]. It seems that extreme deficiency in DC2 is well-compensated for by other desmocollins in the skin (i.e., DC1 and DC3). Nonetheless, more variants should be investigated to draw decisive conclusions. Meanwhile, with the limited functional evidence in mind, patients with DC2 reduction might benefit from native protein-increasing therapeutic strategies. Taken into account, over-administration may be detrimental to humans, as *DSC2* overexpression caused severe cardiac dysfunction in mice [93].

## 3. Proteins Involved in Intermediate Filament Regulation

The intermediate filament network is part of the cell scaffold that contains multiple keratins (*KRT1, KRT10, KRT5, KRT14* etc.) in keratinocytes and desmin (*DES*) in cardiomyocytes [94] (Figure 1). Genetic variants in genes encoding intermediate filaments may severely harm the integrity of a tissue, as it affects resistance to cellular stretch [95]. *DES* variants may cause striated muscle disease, and 70% of all reported *DES* variants have been implicated in cardiac disease [96,97]. Meanwhile, in the epidermis, different keratins are expressed solely in a differentiation-dependent manner. Basal keratinocytes mostly contain keratin 5 (*KRT5*) and keratin 14 (*KRT14*), and variants in both have been associated with EBS [4,5]. In addition, other gene variants that relate to basal keratin turnover, have been implicated in this disease [4]. Since desmin is restricted to striated muscle and keratins to epithelial cells, variants located within these genes do not cause a cardiocutaneous phenotype. However, recent findings have shown that the skin and heart have ubiquitin ligase KLHL24 as an analogous regulator of intermediate filaments in common [12].

### 3.1. Reported KLHL24 Variants

The gene *KLHL24* encodes the 68 kDa kelch-like protein 24, referred to as KLHL24 [98]. KLHL24 is part of the kelch-like protein family, which contains 42 identified KLHLs. KLHLs play a key role in ubiquitinating many different substrates, which is mediated by the activity of E3-ligases [99]. From the N- to the C-terminal side, KLHL24 contains a BTB-domain, BACK-domain and six kelch-repeats (Figure 5). KLHL24, specifically, was implicated in the turnover of keratins via ubiquitination and subsequent proteasomal degradation [100]. The ClinVar reported 78 variants in *KLHL24*. Of these, 37 are claimed (likely) pathogenic; 34 (likely) benign and seven as variants of unknown significance. Eight variants with functional evidence were found (see Figure 5 and full report in Table 4). Furthermore, at least one variant has been translated into a transgenic mouse model, and a zebrafish knockout model has been developed. The predictions on protein level were probably correct in 1/8 variants and incorrect in 6/8 variants, while the evidence of one variant is somewhat inconclusive as to whether protein is produced or not.

#### 3.1.1. KLHL24 Variants Causing KLHL24 Reduction

Two recessive variants (Table 4) that cause complete loss of KLHL24 function have been reported in patients. Both the homozygous missense (c.917G > A) and nonsense (c.1048G > T) variant resulted in HCM in patients, fitting with the higher desmin levels found in their myocardial biopsies [107]. Moreover, heterozygous parents did not display signs of cardiac disease. Furthermore, no skin phenotype was observed in heterozygous or homozygous state. Absence of KLHL24 was also linked to lethal arrhythmias in zebrafish [107]. Currently, no other investigated variants associated KLHL24 deficiency (≤50% protein) with disease, suggesting that pathological threshold levels of KLHL24 in the heart may lie somewhere between 0–50% of wildtype expression. In the skin, reduction in KLHL24 apparently does not cause a skin (fragility) phenotype [107].

#### 3.1.2. KLHL24 Variants Causing an Altered KLHL24 Protein

Five heterozygous start-codon variants (Table 4) c.1A > T, c.1A > G, c.2T > C, c.3G > A and c.3G > T, and one predicted nonsense variant c.22A > T were associated with congenital aplasia cutis, skin fragility, PPK, alopecia and DCM [12,98,100,101,105,107,108,109]. Disregarding the inaccurate predictions, all six variants were pathogenic and resulted in translation initiation or re-initiation at Met29. Overexpression studies determined that these six variants lead to an N-terminally truncated protein Val2_Met29del, 28 amino acids shorter than the wildtype counterpart [98,100,105]. This truncated protein is less affected by auto-ubiquitination, which results in a longer half-life and disproportionate degradation of keratin 14 [98] in the skin and desmin in the heart [101,105]. In dynamically loaded engineered heart tissues derived from hiPSC from patients (c.1A > G), 10-fold lower levels of desmin were observed which caused all the clinical characteristics of DCM. This result was in line with a reduced expression and phenotype of the explanted heart of one of the patients [101]. Meanwhile, another study reported that KLHL24 is also responsible for the turnover of foetal-like keratins 7,8,17 and 18 [103], while degradation of keratin 14 may be higher in foetal-like hiPSC-derived keratinocytes compared to adult keratinocytes [102]. These data would explain why patients develop congenital aplasia cutis but only mild skin fragility later in life. Furthermore, another study investigating the occurrence of alopecia showed that KLHL24 in skin regulates hair maintenance by mediating the stability of keratin 15. The presence of mutated protein Val2_Met29del disrupted the structure of hair stem follicles, leading to alopecia in mice [104].

#### 3.1.3. Potential Therapeutic Avenues

The abovementioned studies have indicated keratins and desmin as natural targets for KLHL24. This seems to be a tight balance as the gain-of-function *KLHL24* variants cause excessive breakdown of desmin and keratins. These results furthermore indicate that patients displaying variants that result in expression of Val2_Met29del would likely benefit from *KLHL24* RNAi therapies, as this has effectively regained keratin and desmin protein levels in in vitro models [98,100,101,102]. While unlikely to affect the skin, excessive RNAi can be detrimental to cardiac function, as absence of KLHL24 has been associated with HCM. Opposingly, patients lacking KLHL24 could benefit from opposing strategies that attain its expression in the heart, minding that the opposite effect (i.e.,: DCM) can be achieved by administrating too high dosages. Strategies to mitigate all patient’s symptoms thus must be precisely executed. Regarding the expression of KLHL24 in tissues: although mRNA expression of *KLHL24* is high in both the skin and heart, protein levels have nearly been impossible to detect in patient-derived materials. This complicates the analysis of novel variants using Western blotting techniques to assess protein pathology. Functional interpretation must therefore shift towards artificial cell models or protein target quantification (i.e., keratins and desmin). Of the functionally assessed *KLHL24* variants, it is clear that the effect of the variant on the protein is difficult, if not impossible, to predict in silico in the case of *KLHL24*.

## 4. Gap Junction Genes

Gap junctions are channels at cell–cell junctions that function for intercellular communication and exchange of molecules (Figure 1). They are formed by connexins (Cx, *GJA*), a family of at least 20 different transmembrane proteins that can assemble into hemichannels, also called connexomes, clusters of six proteins forming a pore between two adjacent cells [110]. Different kinds of connexins are expressed in the skin and heart [111,112,113]. Gap junctions in the heart, specifically in cardiomyocytes, are located within the intercalated disc, in close proximity to desmosomes. Moreover, connexomes have been found to be in direct communication with desmosomes and are often defective in function and/or structure in a cardiocutaneous syndrome and other cardiac pathologies [37,114,115]. Due to the versatility of the transport function of connexins, variants in these proteins can cause a broad range of distinct diseases. Connexin 43 (Cx43), encoded by *GJA1,* is a connexin expressed in both the skin and heart, and variants in *GJA1* are known to cause a broad range of phenotypes, including PPK, alopecia and cardiac conductive disorders thus capable of causing a cardiocutaneous phenotype.

### 4.1. Reported GJA1 Variants

The gene *GJA1* encodes a 43 kDa connexin (Cx43) protein. The hemichannel pore between adjacent cells is formed by six connexins that form a hexamere. Each individual Cx43 contains four transmembrane domains (TM1-4) linked by a cytoplasmic and two extracellular loops and their N-terminal and C-terminal fragment within the cytoplasm [116,117] (Figure 6). The ClinVar database reported 217 variants in *GJA1*. Of these, 58 were claimed (likely) pathogenic; 45 (likely) benign; 16 show conflicting interpretations and 98 have an unknown significance. We found 36 variants substantiated by experiments containing functional evidence (see Figure 6 and full report in Table 5). Furthermore, at least five variants have been translated into transgenic mouse models. Still, 83% of the reported *GJA1* variants have not been functionally investigated. Since *GJA1* has a single coding exon and can therefore not be spliced, the rate of prediction success is high, and the predictions on protein level were correct in 36/36 of the functionally investigated variants.

#### 4.1.1. GJA1 Variants Causing Cx43 Reduction

In humans, variants causing partial or complete Cx43 depletion have not been functionally investigated. Of note, during development, connexins, derived from other genes than *GJA1*, are important. Therefore, full- and heart-conditional *GJA1* knockout mice die shortly after birth due to major heart defects. Due to the short lifespan of these mice, the effects of total loss of Cx43 in the skin are unclear, especially in later stages of development [162]. These data nonetheless indicate that human variants leading to complete Cx43 depletion are likely not compatible with life.

#### 4.1.2. GJA1 Variants Causing an Altered Cx43 Protein

All 36 functionally investigated variants caused an altered, less functional gap junction, predominately due to heterozygous missense variants (31/36). In addition, four nonsense/nonsense-inducing variants and one in-frame indel variant were investigated. Over 90% of these 36 variants affected the eyes, teeth and fingers, diagnosed as the condition oculodentodigital dysplasia. Almost all of these variants were investigated solely in that context, but they can also cause keratoderma, hair abnormalities (alopecia) and cardiac problems, although they are not always specified/reported. While most in silico algorithms accurately assessed the deleterious effect of variants, contradictory predictions still occurred. Meanwhile, in at least 14 out of the 36 investigated variants, researchers clearly demonstrated the interference of mutants with normal gap junction formation, where both mutant and wildtype Cx43 integrated into hemichannels. In the skin, Cx43 plays a key role in wound healing and intercellular communication, the latter being critical for cell differentiation, proliferation, migration and ion transport, and most variants caused serious harm to these processes [13,119,141,154]. The cardiac effects of *GJA1* variants in humans have not been extensively researched. However, they have been associated with arrhythmic sudden death [130,131,163,164], DCM [164] and congenital heart malformations [116,125,149,165].

#### 4.1.3. Potential Therapeutic Avenues

Dominant-negative missense variants, causing detrimental amino acid changes in Cx43, have profound physiological consequences, regardless of the location within the protein. In almost half of the investigated variants, the mutant protein clearly disrupted the formation of normal gap junctions. It is likely that the others caused a similar protein pathology as well. Meanwhile, the pathogenicity prediction of *GJA1* variants through in silico algorithms were closer to accuracy than the other cardiocutaneous genes. Cx43, namely, highly resembles 20 other human connexins, and most amino acids in Cx43 are also highly conserved among species, which makes *GJA1* an ideal candidate gene for in silico predictions. Unfortunately, almost all variants with functional evidence have been studied mainly in relation to oculodentodigital dysplasia [117]. Due to the versatility of Cx43 function, further studies should widen the scope in order to establish the full role of this protein in skin and heart disease. Nonetheless, strategies to ease the overall phenotype of patients should be directed towards conserving the functional hexameric complex structure of hemichannels at all costs. Patients may therefore benefit from treatments that focus on eliminating mutant proteins (i.e., RNAi, CRISPR).

## 5. Discussion

A substantial number of genetic variants in the genes *DSP*, *JUP*, *DSC2*, *KLHL24* and *GJA1* have been reported to underly skin and/or cardiac disease. In this review, we show that only few of them have been functionally assessed, and when they have, the functional data show that the effect of the variant on the protein and its function are quite often not as predicted by in silico prediction programs. The conclusions that can be drawn from functional evidence are important in light of genotype–phenotype correlations (and prognosis) and of potential targeted therapies. The data of this review show that there is a need for functional studies of variants, instead of relying on in silico prediction. In addition, although data on the protein pathology of variants has begun to shape the understanding of disease, there is still a long way to go. Several state-of-the-art in vitro approaches are now available that can significantly enhance insight into the genotype–phenotype correlation and the functional effects of the variants involved in cardiocutaneous syndromes in order to understand and subsequently develop therapies for the resulting phenotypes.

The correct interpretation of variants is an important step towards understanding cardiocutaneous disease mechanisms: a factor that will provide better recommendations for variant carriers and ultimately therapeutic interventions for the patients that are at risk. However, the overview presented here points out that the functional effect of most variants still needs to be established. Moreover, predictions frequently contradicted the functional evidence, exemplified by nonsense variants that did not necessarily reflect absence of protein, NMD that was not always acting according to current assumptions, and similarly altered proteins with different deleterious effect on tissue homeostasis.

In the meantime, the interpretation of variants can be strengthened by applying a few minor additions/alterations to guidelines that assess functional studies. The ACMG guidelines, published in 2015 [14] and following recommendations from the Sequence Variant Interpretation (SVI) working group [166], classify functional studies into different categories of evidence from low to high such as: ‘supporting’, ‘moderate’ or ‘strong’. In this classification, functional assays derived from one variant carrier (patient) compared to one control individual easily fall into the category ‘supporting’. Likewise, experiments that use artificial cell models and variant transfection methods can reach a similar status. Meanwhile, in order to reach a ‘moderate’ line of evidence, studies need to include data of more than ten control individuals, in addition to data from a carrier with a known benign and known pathogenic variant. Using these criteria, the functional evidence of variants included in this overview were given a ‘supporting’ status at maximum, while they provide a wealth of information. Using current guidelines, the functional evidence of studies could not be further differentiated, even though clear distinctions can be made. The following additions/alterations to the guidelines can provide aid. For instance, isogenic controls or variant-targeting interventions that rescue the cellular phenotype are not yet regarded as a criterion in the guidelines, but they do provide significant strength to the line of evidence. These experiments remove genetic background in a more efficient way than inclusion of multiple control individuals. In addition, direct patient or patient-derived material provides stronger lines of evidence than artificial cell models, as the latter do not necessarily reflect the in vivo regulation of gene expression, mRNA splicing and protein quality-control mechanisms [25]. Finally, as the effect of genetic variants may differ per tissue, studies that include multiple cell types provide better insight into the patient as a whole and therefore also provide a stronger line of evidence.

This overview of functional evidence of desmosomal gene variants also points out that variants may exert their effect by either protein reduction or by altered protein function, and that the understanding hereof is eventually critical to the development of therapeutic interventions. For instance, KLHL24 protein activity, whether through loss or gain of function, causes a strong dose-dependent phenotype in the heart, which seems directly related to expression of its target desmin [101,107]. The relationship to keratins in the skin is less straightforward and requires more cohesive data that may now be emerging with new studies [102,103,104]. Functional studies combined also suggest that the phenotype associated with DP reduction is both dose- and organ-dependent in nature, which is important knowledge for future interventions. DC2 deficiency, however, may mostly be related to dose-dependent heart disease, yet sufficient data are lacking. Meanwhile, the phenotype associated with PG reduction is highly organ-dependent and strongly correlated with skin, but not heart disease. More supporting studies are needed to strengthen these findings. In contrast, proteins with an altered function were mostly shown to be erratic and unpredictable in nature. Variants in *GJA1* showed most coherence, as expression of each altered yet stable protein will ultimately interfere with the formation of normal gap junctions. Furthermore, variants that alter the function of DP, PG and DC2, were more diverse in phenotypic results. Some general conclusions were drawn, but most require substantially more supporting evidence. A weighted summary of the functional evidence of variants to current clinical variant databases could aid in gaining imminent clarity. More importantly, supporting studies on the protein pathology of the remaining >95% of variants that have not been functionally evaluated will unravel genotype–phenotype correlations in the near future.

With emerging novel techniques and improvements in in vitro models, much more is to be gained from functional studies. Three-dimensional skin models can now be implied, which should significantly reduce the discrepancy in results observed from ex vivo skin biopsies and biopsy-derived monolayer cultures of keratinocytes. In addition, differentiation protocols of patient hiPSC-derived keratinocytes are emerging [167], which even allow for the formation of complete hair-bearing skin [168]. With these techniques, the whole epidermis can be engineered and mechanical abrasions that provoke disease can be mimicked in vitro. In addition, significant advances have been made to study heart disease. Differentiation protocols of patient hiPSC-derived cardiomyocytes are already highly advanced and broadly applied. In addition, engineered heart tissues can now be made [169] and stimulated to allow for highly aligned cardiac fibres [170]. They can also undergo pre-and afterload in a dynamic fashion, thereby increasing the fractional shortening up to 20–30%, like in the in vivo human heart [29]. In conclusion, more emphasis on functional evaluation in variant prediction guidelines and establishment of functional protein pathologies resulting from genetic variants and state-of-the art techniques to do so can significantly enhance insight into the pathophysiology of cardiocutaneous diseases, which will be paramount to the development of therapies.

## 6. Materials and Methods

In August 2022, the ClinVar database was used to assess the current number of reported variants [171]. From 1995 to August 2022, PubMed entries on genes *DSP* (DSP; DP; desmoplakin {P15924}), *JUP* (JUP; plakoglobin; ɣ-catenin; PG {P14923}), *DSC2* (DSC2; desmocollin-2 {Q02487}), *GJA1* (GJA1; Cx43; Connexin-43; gap junction protein alpha 1 {P17302}) and *KLHL24* (KLHL24; Kelch-like family member 24; DRE1; KRIP6) were evaluated. PRISMA guidelines were adhered to, and the results contained the following inclusion criteria: PubMed articles containing disease variants that published quantitative immuno-blot and/or mRNA levels, obtained from patient-derived cell/tissue sources and/or transfection studies, were included. In addition, transgenic, homozygous or heterozygous animal knockout models were included, if they reflected the pathophysiology of reported human variant(s). We separated the functionally investigated variants into a ‘protein reducing’ or ‘altered protein function’ category. In the case of variants that cause protein reduction: in the heterozygous state, protein expression from one allele is lost, resulting in an expected 50% reduction of total protein. Meanwhile, in the homozygous state, this reduction is expected to be 100%. As these numbers may vary, the amount of protein reduction is summarized for each variant and potential carrier types. In the case of variants that result in a protein with an altered function: these are regarded as expressed within the cell, but they may or may not give a normal dosage. In the heterozygous state, an expected 50% of the total protein is native, while the other 50% of total protein is mutated. Homozygous carriers, however, have no native protein expression. As these numbers may vary, the size and abundancy of the expressed mutant are summarized for each variant and potential carrier type. Moreover, all the included variants were reclassified by us, according to ACMG-AMP guidelines [14] and following recommendations [166,172,173,174]. In addition, we also reported the predicted variant pathogenicity through use of in silico algorithms and the predicted variant outcome according to the following general considerations: Missense variants, which generate protein variants with a single amino acid variation, may induce drastic structural alterations or may induce structural alterations able to perturb binding interfaces. Missense variants normally do not affect protein production processes, but proteins with single amino acid variation may be degraded if severely unstable. The pathogenicity of missense variants was predicted by us using SIFT [175], PolyPhen-2 [176] and MutPred2 [177]. In-frame insertions and deletions (indel) cause an addition and/or deletion of certain amino acids in the protein. Like missense variants, these variants normally do not cause absence of protein, but proteins with in-frame indels may be degraded if severely unstable. The pathogenicity of in-frame indels was predicted by us using MutPred-Indel [178]. Nonsense variants (i.e., stop-gain variants, out-of-frame indels leading to premature downstream stop-gain codons) introduce a premature termination codon (PTC). On occasion, nonsense variants in the first exon may cause an N-terminal truncated protein, due to translation re-initiation (if another downstream start-codon is activated). Meanwhile, nonsense variants in the terminal exon normally result in a C-terminal truncated protein. Both types of truncations are expected to be expressed in the cell, but these proteins may be degraded if severely unstable. Gene transcripts with nonsense variants located between exon 1 and the terminal exon are predicted to be broken down by nonsense-mediated decay (NMD). In mammals, this system is predicted to operate when at least one intron downstream of the premature termination codon is present [179]. It is thus predicted that the truncated protein is not expressed in the cell. The pathogenicity of nonsense variants was predicted by us using MutPred-LOF software [180]. Finally, splice-site variants can affect the splicing of introns and exons, and outcomes such as (partial) intron retention or exon skipping may occur. The Human Splicing Finder and MaxEntScan software were used by us to predict the alternative splicing odds. Furthermore, there are no prediction algorithms available for either 5′UTR or 3′URT variants. Finally, the functional evidence of each variant was compared to the in silico predictions, and this was categorized as a ‘match’, if the predictions were on par with the functional evidence observed in patient-derived cells/tissues and or transfection/animal studies; ‘probably match’ if the functional predictions were on par with the functional evidence but decisive proof of the latter was incomplete; ‘mismatch’ if the functional predictions were not on par with the functional evidence; and ‘unclear’ if the functional evidence itself was inconclusive or contradictory.

## Figures and Tables

**Figure 1 ijms-23-10765-f001:**
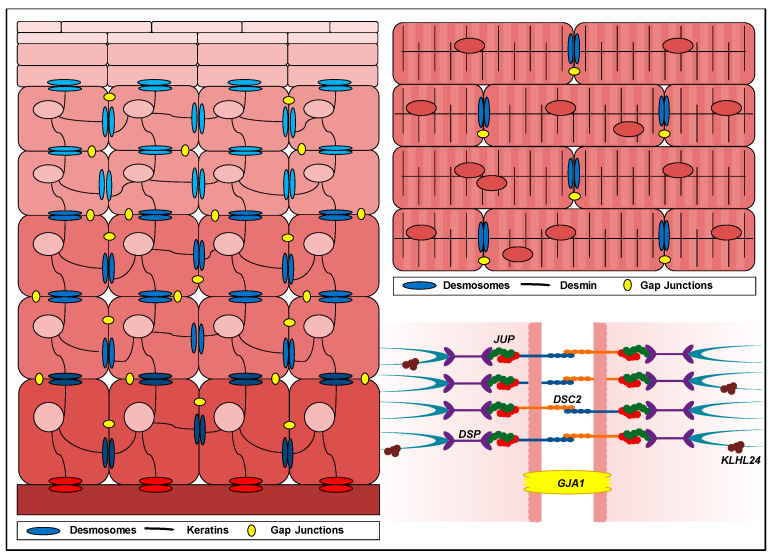
Schematic overview depicting the location of proteins causative for a cardiocutaneous syndrome. The left panel depicts continuously differentiating keratinocytes in the skin, migrating from the basement membrane towards the stratum corneum, where the terminally differentiated keratinocytes are localized. Keratinocytes are interconnected by gap junctions (in yellow) and desmosomes (in blue) that anchor the keratin intermediate filament network. Hemidesmosomes (in red) connect the basal keratinocytes to the underlying dermis. The upper right panel depicts a zoomed-in fraction of the myocardial wall, where cardiomyocytes are interconnected by gap junctions and desmosomes at the intercalated disc. The lower right panel depicts a plasma membrane between two adjacent cells containing a gap junction, connexin 43 (gene *GJA1,* protein Cx43) and a desmosomal junction, consisting of desmoplakin (gene *DSP,* protein DP), plakoglobin (gene *JUP,* protein PG), desmocollins (subtype 2; gene *DSC2*, protein DC2) desmogleins (blue protein structures) and plakophilins (red protein structures). Intermediate filaments (turquoise protein structures) adhere to the desmosomal junction. Kelch-like protein 24 (gene and protein *KLHL24*) mostly accumulates near the plasma membrane.

**Figure 2 ijms-23-10765-f002:**
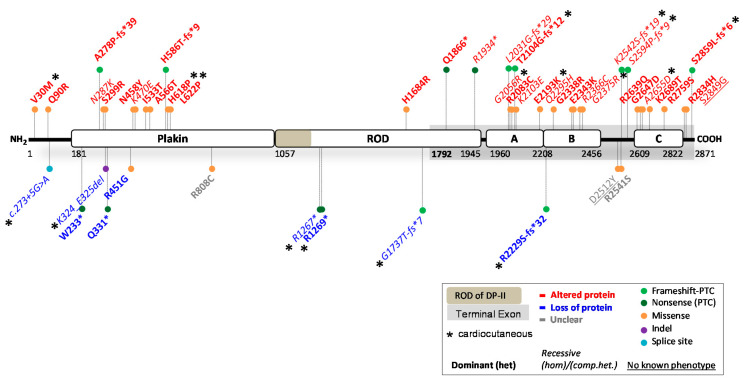
Location of functionally investigated *DSP* variants.

**Figure 3 ijms-23-10765-f003:**
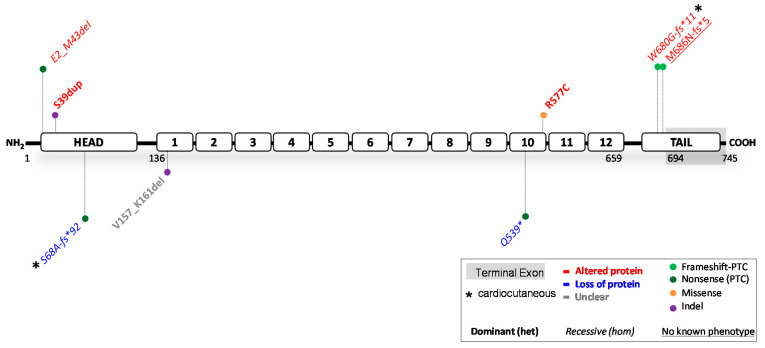
Location of functionally investigated *JUP* variants.

**Figure 4 ijms-23-10765-f004:**
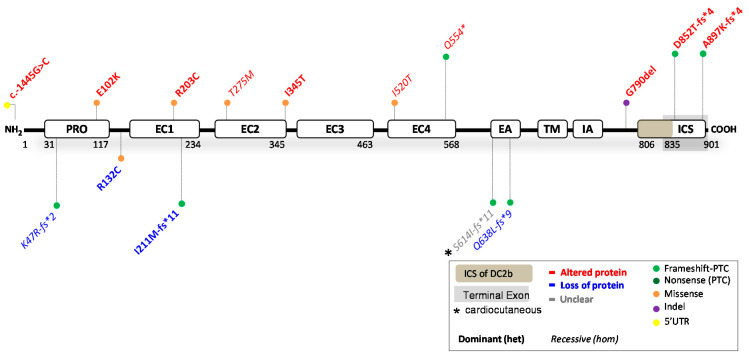
Location of functionally investigated *DSC2* variants.

**Figure 5 ijms-23-10765-f005:**
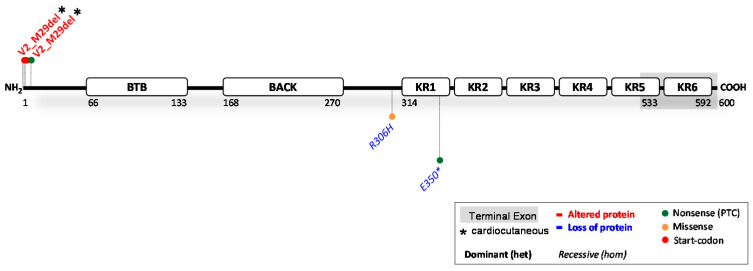
Location of functionally investigated *KLHL24* variants.

**Figure 6 ijms-23-10765-f006:**
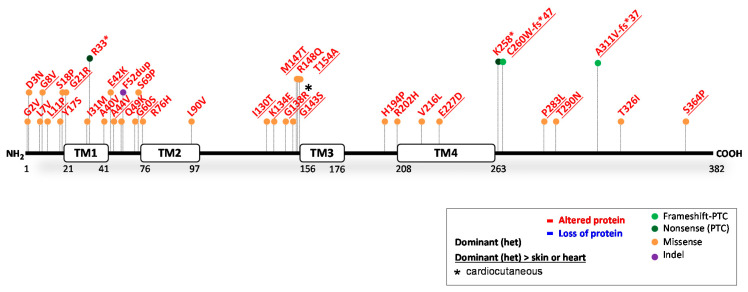
Location of functionally investigated *GJA1* variants.

**Table 1 ijms-23-10765-t001:** Functionally analysed *DSP* variants.

HGVS Nomenclature (DNA)	HGVS Nomenclature (Protein)	Protein Domain	ACMG Class	In Silico Predictions	Functional mRNA and Protein Studies	Biological Effect	Prediction: Functional	Skin	Heart
c.88G > A	p.(Val30Met)	N-terminus	B	>Protein expressed -PolyPhen-2> Benign (0.000) -SIFT> NOT tolerated -MutPred2> Benign (0.092)	*Altered DP function;* Mutant DP protein expressed, normal size and amount (WB) [24,25,26].	Binding to PG abolished (Co-IP); DP localization in cytoplasm (transfection) [24]; DP normal in myocardial and epidermal tissue. Exhibit weaker binding to iASPP (transfection) [26]. Mouse *DSP*^WT/88G > A^ [24].	Match	(het) PPK; (het) WH^2^	(het) ACM
c.269A > G	p.(Gln90Arg)	N-terminus	B	>Protein expressed -PolyPhen-2> Probably damaging (0.967) -SIFT> NOT tolerated -MutPred2> Pathogenic (0.757)	*Altered DP function;* Mutant DP protein expressed, normal size and amount (WB) [24,26].	Binding to PG abolished (Co-IP); DP localization to cytoplasm (transfection) [24]. Mouse *DSP*^WT/269A > G^ [24]	Match	n.s.	(het) ACM
c.273 + 5G > A	Multiple splice products	Intron splice site (N-terminus)	US	-Human Splicing Finder> Broken WT donor site -MaxEntScan> Alteration of WT donor site, probably affecting splicing > Altered splicing, out of frame> PTC > NMD	*Partial loss of DP:* 20% less DP product on WB. No alternatively spliced transcripts discovered in patient-derived cells ([27,28], but did so in in vitro splicing assay (transfection). However, not functional [28].	-In combination with c.6687del> Reduced DP protein on blot and staining in explanted heart and hiPSC-CMs and primary KCs [28,29]. -Dislocation of DP after 2D mechanical stretch; resulted in reduced count and density of desmosomes (EM) in dynamic EHTs leading to lower force and stress [29].	Partial match, normal splicing left.	(comp.het) PPK; WH with *DSP:* c.6687delA	(comp.het) ACM/ NCCM :with *DSP:* c.6687delA
c.699G > A	p.(Trp233*)	Plakin-domain	P	>PTC > NMD, no protein -MutPred-LOF> Borderline pathogenic (0.55385)	*Partial loss of DP*; Mutant RNA not detected in patient cells. Mutant DP is unstable (transfection-WB) [24].	Perinuclear aggregates of DP (transfection IF) [24].	Partial match, but not with transfection IF/WB	n.s.	(het) ACM
c.832del	p.(Ala278Pro fs*39)	Plakin-domain	P	>PTC > NMD, no protein	*Altered DP function;* Truncated DP normally expressed and protein runs at 60 kDa (315 aa) (transfection-WB) -Leads to truncated *DSP* mRNA, also indicating that mRNA translation following the truncation was completely impaired.	c.832del overexpression led to upregulation of PG and downregulation of β-catenin in the nuclei, without affecting their expression in the cytoplasm (transfection) [30].	Mismatch	n.s.	(het) ACM
c.861T > G	p.(Asn287Lys)	Plakin-domain	LP	>Protein expressed -PolyPhen-2> Probably damaging (0.997) -SIFT> NOT tolerated -MutPred2> Pathogenic (0.699)	*Altered DP function;* Mutant DP expressed [31].	Aberrant DP and Cx43 localization (transfection-IF) [31].	Match	(hom)PPK; (hom)WH; (hom)EBS	n.o.
c.897C > G	p.(Ser299Arg)	Plakin-domain	LP	>Protein expressed -PolyPhen-2> Probably damaging (0.999) -SIFT> NOT tolerated -MutPred2> Pathogenic (0.672)	*Altered DP function;* Mutant DP expressed [26].	Exhibit weaker binding to iASPP = desmosome regulator (transfection) [26].	Match	n.s.	(het) ACM
c.939 + 1G > A	p.(Gln331*)	Donor site intron 7 (Plakin-domain)	P	>PTC > NMD, no protein	*Partial loss of DP;* Absence of detection of mRNA in multiple patient KCs, reported by two studies, suggests efficient NMD [32,33]. Only 20% DPI and 50% DPII is left on WB [34].	-Major abnormalities in the spinous layer of the epidermis. The intercellular space is widened and KCs contain abnormal cytoplasmic densities [33]. -Small desmosomes and fewer in number; perinuclear keratin distribution ^7^. - DC3 seems reduced on WB; volume densities of desmosomal proteins seem different from control [34].	Match	(het) PPK	n.r.
c.969_974del	p.(Lys324_Glu325del)	Plakin-domain	LP	>Protein expressed -MutPred-Indel> Benign (0.19566)	*Partial loss of DP*; Reduced expression of both native DP isoforms in cytoskeletal and cytoplasmic protein fractions (WB) [25], suggesting instable protein> incomplete degradation.	DP expression was significantly reduced in myocardial tissue and epidermal biopsies (IF) [25].	Mismatch	(hom)PPK; (hom)WH	(hom) ACM, bi-ventricular
c.1348C > G	p.(Arg451Gly)	Plakin-domain	US	>Protein expressed -PolyPhen-2> Probably damaging (1.000) -SIFT> NOT tolerated -MutPred2> Pathogenic (0.756)	*Partial loss of DP;* 50% reduced DPI&II protein in EHTs (WB) [35]. -mRNA levels of *DSP* not reduced compared to WT [35]. > instable protein degradation.	50% reduced DP signal and 70% reduced Cx43 in myocardial tissues (IF); Proteolytic degradation by calpain, leading to DP insufficiency [35].	Mismatch	n.r.	(het) ACM, bi-ventricular
c.1372A > T	p.(Asn458Tyr)	Plakin-domain	US	>Protein expressed -PolyPhen-2> Possibly damaging (0.939) -SIFT> Tolerated -MutPred2> Benign (0.323)	*Altered DP function;* Mutant DP expressed [31].	Altered EB1 binding and Cx43 localization (transfection IF) [31].	Match	n.o.	(het) ACM
c.1408A > G	p.(Lys470Glu)	Plakin-domain	US	>Protein expressed -PolyPhen-2> Benign (0.082) -SIFT> Tolerated -MutPred2> Benign (0.408)	*Altered DP function;* Conformational alternation, but overall folded structure of DP is remained [36]. Mutant DP expressed (WB) [31].	Mutant is incorporated into the desmosome [25].	Match	n.s./n.o.	(hom) ACM
c.1598T > C	p.(Ile533Thr)	Plakin-domain	US	>Protein expressed -PolyPhen-2> Probably damaging (0.998) -SIFT> NOT tolerated -MutPred2> Benign (0.442)	*Altered DP function;* Mutant DP expressed (WB) [31].	Altered EB1 binding and Cx43 localization (transfection IF) [31].	Match	n.o.	(het) ACM
c.1696G > A	p.(Ala566Thr)	Plakin-domain	US	>Protein expressed -PolyPhen-2> Benign (0.007) -SIFT> Tolerated -MutPred2> Benign (0.153)	*Altered DP function;* Mutant DP expressed (WB) [31].	Mutant is incorporated into the desmosome [25].	Match	n.o.	(het) ACM
c.1853A > C	p.(His618Pro)	Plakin-domain	LP	>Protein expressed -PolyPhen-2> Possibly damaging (0.602) -SIFT> NOT tolerated -MutPred2> Benign (0.540)	*Altered DP function;* Mutant DP expressed (WB) [37].	Mutant localizes to membrane, affected Cx43 localization (transfection studies/skin biopsies). Desmosome aggregation [37].	Match	(het) PPK; (het) WH; (het) EBS	(het) CM
c.1865T > C	p.(Leu622Pro)	Plakin-domain	LP	>Protein expressed -PolyPhen-2> Probably damaging (0.998) -SIFT> NOT tolerated -MutPred2> Pathogenic (0.828)	*Altered DP function;* Mutant DP expressed (WB) [37].	Mutant localizes to membrane, affected Cx43 localization (transfection studies/skin biopsies). Desmosome aggregation [37].	Match	(het) PPK; (het) WH; (het) EBS	(het) CM
c.1755dup	p.(His586Thr fs*9)	Plakin domain	P	>PTC > NMD, no protein	*Altered DP function;* Truncated DP protein, (65 kDa) (WB), truncation of ROD and C-terminus [38].	n.r.	Mismatch	n.s.	(het) ACM, LV mostly
c.2422C > T	p.(Arg808Cys)	Plakin-domain	US	>Protein expressed -PolyPhen-2> Benign (0.047) -SIFT> NOT tolerated -MutPred2> Benign (0.409)	*Unclear;* Conformational alteration (transfection), but overall folded structure of DP is remained [36]. Needs further confirmation in patient cells, whether expressed or not.	n.r.	Unclear	n.s.	(het) ACM
c.3799C > T	p.(Arg1267*)	ROD domain (DPI)	P	>PTC > NMD, no protein -MutPred-LOF> Borderline pathogenic (0.5552)	*Partial loss of DPI;* Instable mutant DPI protein (NMD?) [39]. -Highly reduced *DSP* mRNA expression (NMD?) [39]. -Complete loss of DPI in patient skin, DPII has normal expression as expected (WB) [39].	n.r.	Match	(hom)PPK epidermolytic; (hom)WH	(hom) ACM/ DCM
c.3805C > T	p.(Arg1269*)	ROD domain (DPI)	P	>PTC > NMD, no protein -MutPred-LOF> Borderline pathogenic (0.55487)	*Partial loss of DPI;* Broken down by NMD. DPI/DPI-II protein ratios lower in variant carriers compared with WT individuals. -DPI/DPII expression ratio reduced by 28% in mutant cells. 15-fold lower mutant than WT [25].	Decreased DP expression in endomyocardial biopsies. DPI deficiency (IF) [25].	Match	(het) PPK; (het) WH	(het) DCM, bi-ventricular
c.5051A > G	p.(His1684Arg)	ROD domain (DPI)	US	>Protein expressed -PolyPhen-2> Possibly damaging (0.956) -SIFT> NOT tolerated -MutPred2> Benign (0.256)	*Altered DPI function:* No effect on amount or size of DPI protein on WB [40]. DPII should not be affected.	Affects action potential and duration; multiple ion channel dysfunction in hiPSC-CMs [40].	Match	n.r.	(het) CM, conduction disease
c.5208_5209del	p.(Gly1737Thrfs*7)	ROD domain (DPI)	P	>PTC > NMD, no protein	*Partial loss of DPI/Unclear?* Truncated DPI protein predicted to run at similar height as DPII, yet no increase in this band was observed in skin biopsies (WB) [41]. DPII should not be affected, but data are unclear.	n.r.	Unclear	(hom) PPK acantholytic; (hom)WH;	(hom) NCCM, bi-ventricular, severe
c.5596C > T	p.(Gln1866*)	ROD domain (DPI)	LP	>PTC > terminal exon, NOT NMD > protein expressed	*Altered DPI function;* Truncated DPI protein (160 kDa) observed in skin biopsies [42]. DPII should not be affected.	n.r.	Match	n.s.	(het) ACM, LV dilation
c.5800C > T	p.(Arg1934*)	ROD domain (DPI)	LP	>PTC > terminal exon, NOT NMD > protein expressed	*Altered DPI function;* Truncated DPI protein (243 kDa) (WB) [43]. DPII should not be affected. -Aberrant mRNA transcripts. Not NMD.	Stable expressed DP protein, which is recruited into desmosomes, although more punctate staining was observed (IF) [43].	Match	(comp.het) (lethal) EBS, PPK and WH, with *DSP:* c.6091_6092del [43]	n.r.
c.6091_6092del	p.(Leu2031Glyfs*29)	PRD (A domain)	LP	>PTC > terminal exon, NOT NMD > protein expressed	*Altered DP function;* Truncated DP-I protein (228 kDa) (WB) [43]. Not clear what happens with DPII. -Aberrant mRNA transcripts. Not NMD.	Stable expressed DP protein, which is recruited in desmosomes, although more punctate staining was observed (IF) [43].	Match	(comp.het) (lethal) EBS, PPK; and WH, with *DSP*: c.5800C > T [43]	n.r.
c.6166G > C	p.(Gly2056Arg)	PRD (A domain)	US	>Protein expressed -PolyPhen-2> Probably damaging (1.000) -SIFT> NOT tolerated -MutPred2> Pathogenic (0.872)	*Altered DP function*; Expressed in insoluble fraction of bacterial cells (transfection WB) [44]. Low expression in HeLa cells. Likely expressed mutant.	n.r.	Probable match	(hom) PPK	(hom) ACM, LV involvement
c.6247C > T	p.(Arg2083Cys)	PRD (A domain)	US	>Protein expressed -PolyPhen-2> Probably damaging (1.000) -SIFT> NOT tolerated -MutPred2> Benign (0.443)	*Altered DP function*; Expressed in soluble fraction of bacterial cells (transfection WB), thus correctly folded [44]. Likely expressed mutant, needs confirmation in patient cells.	n.r.	Probable match	n.r.	(het) LQTS
c.6307A > G	p.(Lys2103Glu)	PRD (A domain)	US	>Protein expressed -PolyPhen-2> Possibly damaging (0.860) -SIFT> Tolerated -MutPred2> Benign (0.417)	*Altered DP function*; Expressed in soluble fraction of bacterial cell transfection, thus correctly folded (WB) [44]. Likely expressed mutant, needs confirmation in patient cells.	n.r.	Probable match	n.r.	(het) DCM
c.6310del	p.(Thr2104Glnfs*12)	PRD (A domain)	LP	>PTC > terminal exon, NOT NMD > protein expressed	*Altered DP function;* Several truncated DP proteins shown on WB, but mutant is predicted to be 238 kDa [45]	Fibrosis and fat deposition in the heart with reduction in Cx43, disorganized IDs, but staining of DP, PG and DG2 seemed normal; severe reduction of DPI&II on IF ex vivo skin. β-catenin expression was also reduced on IF in skin [45].	Match	(comp.het) EBS, PPK and WH: with *DSP:* c.7964C > A	(comp.het) Bi-ventricular CM: with *DSP:* c.7964C > A
c.6577G > A	p.(Glu2193Lys)	PRD (A domain)	US	>Protein expressed -PolyPhen-2> Possibly damaging (0.950) -SIFT> Tolerated -MutPred2> Benign (0.346)	*Altered DP function*; Expressed in insoluble fraction in bacterial cells (transfection WB) [44]. Likely expressed mutant, needs confirmation in patient cells.	n.r.	Probable match	(comp.het) Alopecia PPK, with *DSP*:c.7567delAAGA	(comp.het) DCM, with *DSP*:c.7567delAAGA
c.6687del	p.(Arg2229Serfs*32)	PRD (B domain)	LP	>PTC > terminal exon, NOT NMD > protein expressed	*Partial loss of DP:* NMD of product (WB, NMD inhibitor exp.), 50% reduced protein levels [28,29]. -mRNA 50% reduced [29].	-Reduced DP protein on blot and staining in explanted heart, hiPSC-CMs and primary KCs [28,29]. -Mislocalisation of DP after 2D mechanical stretch; in combination with c.273 + 5G > A resulted in reduced count and density of desmosomes in hiPSC-derived dynamic EHTs leading to lower force and stress [29] -Faster differentiation observed in primary KCs of patients. Mechanical stretch provoked cell-contact defects [28].	Mismatch NMD active in terminal exon!	(het)PPK; (comp.het)WH: with *DSP*: c.273 + 5G > A	Lethal ACM/ NCCM (comp.het) ACM/ NCCM (bi-ventricular) (het)
c.6885A > T	p.(Gln2295His)	PRD (B domain)	US	>Protein expressed -PolyPhen-2> Probably damaging (0.999) -SIFT> NOT tolerated -MutPred2> Pathogenic (0.833)	*Altered DP function;* Likely truncated DP protein expressed.	Severe binding deficiency with intermediate filaments (transfection IF) [46].	Probable match	n.r.	(hom) DCM
c.7012G > A	p.(Gly2338Arg)	PRD (B domain)	US	>Protein expressed -PolyPhen-2> Probably damaging (1.000) -SIFT> NOT tolerated -MutPred2> Pathogenic (0.910)	*Altered DP function*; Insoluble fraction in bacterial cell transfection (WB) [44]. Likely expressed mutant, needs confirmation in patient cells.	n.r.	Probable match	n.r.	(het) CM
c.7027G > A	p.(Glu2343Lys)	PRD (B domain)	US	>Protein expressed -PolyPhen-2> Benign (0.077) -SIFT> Tolerated -MutPred2> Benign (0.386)	*Altered DP function;* Likely truncated DP protein expressed. Soluble fraction in bacterial cell transfection, thus correctly folded (WB) [44].	Altered binding with vimentin and keratin8/18 (transfection IF) [46].	Probable match	n.r.	(het) ACM ACM, with *PKP2*:c.1468C > T
c.7096C > T	p.(Arg2366Cys)	PRD (B domain)	LP	>Protein expressed -PolyPhen-2> Probably damaging (0.980) -SIFT> NOT tolerated -MutPred2> Benign (0.622)	*Altered DP function;* Likely truncated DP protein expressed [46]. Soluble fraction of bacterial cell transfection (WB) [44]. High expression in HeLa cells. Needs confirmation in patient cells.	Severe binding deficiency with intermediate filaments (transfection IF) [46]. No binding deficiency with vimentin (IF transfection) [44].	Probable match	(hom)EBS; (hom)PPK; (hom)WH	n.r.
c.7123G > C	p.(Gly2375Arg)	PRD (B domain)	US	>Protein expressed -PolyPhen-2> Probably damaging (1.000) -SIFT> NOT tolerated -MutPred2> Pathogenic (0.938)	*Altered DP function;* Truncated DP protein expressed [47]. Insoluble fraction of bacterial cell transfection (WB) [44].	Co-alignment with IFs severely affected. Diffuse cytosolic distributed [47]. Targeting to IFs affected (transfection-IF) [44].	Match	(hom)EBS; (hom)PPK; (hom)WH	(hom)ACM
c.7534G > T	p.(Asp2512Tyr)	Linkers	US	>Protein expressed -PolyPhen-2> Probably damaging (0.998) -SIFT> NOT tolerated -MutPred2> Pathogenic (0.825)	*Unclear;* Likely truncated DP protein expressed. Needs further confirmation in patient cells.	No binding deficiency with IFs (transfection-IF) [46].	Unclear	n.o.	n.o.
c.7623G > T	p.(Arg2541Ser)	Linkers	US	>Protein expressed -PolyPhen-2> Benign (0.010) -SIFT> NOT tolerated -MutPred2> Benign (0.265)	*Unclear;* Likely truncated DP protein expressed. Needs further confirmation in patient cells.	No binding deficiency with IFs (transfection-IF) [46].	Unclear	n.r.	(het) ACM
c.7623del	p.(Lys2542Serfs*19)	Linkers	LP	>PTC > terminal exon, NOT NMD > protein expressed	*Altered DP function;* Severe reduction of both DPI&II (WB), both truncated proteins detected [25].	Normal DP immunoreactivity in epidermal and myocardial tissue (IF)/or almost no signal depending on homozygous or heterozygous patient [25].	Match	(hom)PPK; (hom)WH	(hom) ACM, bi-ventricular
c.7780del	p.(Ser2594Profs*9)	Linkers	LP	>PTC > terminal exon, NOT NMD > protein expressed	*Altered DP function;* Truncated DP protein, 18 aa downstream of deletion (WB) [1].	Partial disruption with intermediate filament binding (IF) [1]; KCs have alteration in morphology, elasticity, adhesion capabilities and viscoelastic properties [48,49].	Match	(hom)PPK; (hom)WH	(hom) DCM
c.7916G > A	p.(Arg2639Gln)	PRD (C domain)	US	>Protein expressed -PolyPhen-2> Probably damaging (0.978) -SIFT > Tolerated -MutPred2> Benign (0.484)	*Altered DP function;* Likely truncated DP proteins expressed [46]. Expressed in soluble fraction in bacterial cells (transfection WB) [44].	Altered binding with desmin and keratin8/18 (transfection IF) [46]. No binding deficiency with vimentin (IF transfection) [44].	Probable match	n.r.	(het) CM; RV dysfunction
c.7940G > A	p.(Gly2647Asp)	PRD (C domain)	US	>Protein expressed -PolyPhen-2> Probably damaging (0.980) -SIFT> NOT tolerated -MutPred2> Benign (0.619)	*Altered DP function:* Both in insoluble and soluble fraction in bacterial cells (transfection WB) [44]. Likely expressed mutant.	n.r.	Probable match	n.r.	(het) CM
c.7964C > A	p.(Ala2655Asp)	PRD (C domain)	US	>Protein expressed -PolyPhen-2> Probably damaging (0.999) -SIFT> NOT tolerated -MutPred2> Pathogenic (0.754)	*Altered DP function;* Likely truncated DP proteins expressed, as full loss of protein is not expected due to recessive inheritance.	Severe binding deficiency with intermediate filaments (transfection IF) [46].	Probable match	(hom)EBS; (hom)PPK; (hom)WH	(hom) ACM, bi-ventricular
c.8066A > C	p.(Lys2689Thr)	PRD (C domain)	US	>Protein expressed -PolyPhen-2> Probably damaging (1.000) -SIFT > Tolerated -MutPred2> Benign (0.625)	*Altered DP function:* Expressed in soluble fraction in bacterial cells (transfection WB), thus correctly folded [44]. High expression in HeLa cells. Likely expressed mutant. Needs confirmation in patient cells.	No binding deficiency with vimentin (transfection IF) [44].	Probable match	n.r.	(het) ACM
c.8275C > A	p.(Arg2759Ser)	PRD (C domain)	US	>Protein expressed -PolyPhen-2> Probably damaging (1.000) -SIFT > Tolerated -MutPred2> Benign (0.350)	*Altered DP function:* Expressed in soluble fraction in bacterial cells (transfection, thus correctly folded WB) [44]. Likely expressed mutant. Needs confirmation in patient cells.	n.r.	Probable match	n.r.	(het) ACM
c.8501G > A	p.(Arg2834His)	C- terminus	US	>Protein expressed -PolyPhen-2> Probably damaging (0.972) -SIFT> NOT tolerated -MutPred2> Benign (0.189)	*Altered DP function;* C-terminally truncated DP protein (WB) [24].	Aberrant IF localization; DP localization at cell membrane; affects other junctional proteins [24]; Arg2834His blocked the GSK3β phosphorylation cascade and reduced DP–GSK3β interactions in KCs and in hearts of Arg2834His DP mice [18]. Mouse *DSP*^WT/8501G > A^ [24,50,51]	Match	n.s.	(het) ACM
Engineered variant	p.(Ser2849Gly)	C- terminus	n.a.	>Protein expressed -PolyPhen-2> Probably damaging (0.978) -SIFT> NOT tolerated -MutPred2> Benign (0.344)	*Altered DP function;* Mutant DP protein detected (WB), normal size.	Mutant DP exhibits increased anchorage of keratin/desmin [52] filaments and fosters calcium independency [53].	n.a.	unknown	unknown
c.8576_8577del	p.(Ser2859Leufs*6)	C- terminus	LP	>PTC > terminal exon, NOT NMD > protein expressed	*Altered DP function;* Highly reduced mutant DP protein detected in insoluble fraction (WB), none in soluble fraction, but normal size (only 2859 + 6 aa, compared to wildtype 2871 aa)	GSK3β, normally phosphorylates Ser2859Leu, translocated to the soluble fraction of patient extract where its high activity (dephosph). Ser9 was associated with the phosphorylation (Ser33/37-Thr41) and degradation of β-catenin; abolition of β-catenin phosphorylation in the non-soluble fraction was associated with its translocation into CMs nuclei [54].	Match	(hom and het) EBS	(hom and het) ACM

Abbreviations: altered protein function > variant annotated in red; partial loss of protein = variant annotated in blue; unclear > variant annotated in grey; US (uncertain significance); LP (likely pathogenic); P (pathogenic); n.a. (not applicable); n.s. (not specified); aa (amino acids); n.r. (not reported); n.o. (not observed); WT (wildtype); WB (Western blot); IF (immunofluorescence); EM (electron microscopy); NMD (nonsense mediated mRNA decay); fs (frameshift); */PTC (premature termination codon); Co-IP (co-immunoprecipitation); hiPSC (human induced pluripotent stem cells); CMs (cardiomyocytes); EHTs (engineered heart tissues); KCs (keratinocytes); EBS (epidermolysis bullosa simplex); PPK (palmoplantar keratoderma); WH (woolly hair); CM (cardiomyopathy, not further specified); DCM (dilated cardiomyopathy); ACM (arrhythmogenic cardiomyopathy); NCCM (non-compaction cardiomyopathy); LQTS (long QT syndrome); LV (left ventricle); RV (right ventricle); het (heterozygous> phenotype observed in heterozygous carriers); hom (homozygous > phenotype observed in homozygous carriers); comp.het (compound heterozygous > phenotype observed in compound heterozygous carriers).

**Table 2 ijms-23-10765-t002:** Experimental investigation of *JUP* variants.

HGVS Nomenclature (DNA)	HGVS Nomenclature (Protein)	Protein Domain	ACMG Class	In Silico Predictions	Functional mRNA and Protein Studies	Biological Effect	Prediction: Functional	Skin	Heart
c.71C > A	p.(Glu2_Met43del) not p.(Ser24*) as predicted	Head domain	LP	>PTC > NMD, no protein -MutPred-LOF> too short sequence for prediction	*Altered PG function;* Truncated N-terminal protein (lacking the first 42 aa, translation re-initiation Met43) with reduced expression in the skin (WB) [63]. -Similar *JUP* mRNA levels as in control [63].	Reduced PG expression (IF; WB) and disrupted distribution of DP and DG1 (IF) [63].	Mismatch	(hom)EBS; (hom)PPK; (hom)WH	n.o.
c.116_118dup	p.(Ser39dup)	Head domain	US	>Protein expressed -MutPred-Indel> Pathogenic (0.76492)	*Altered PG function:* Mutant PG protein size similar as WT (82 kDa; WB) [64,65]. See comment on biological effect [65].	-Patient heart displayed a decrease in signal of DP, PG and Cx43 (IF ) [64] - Transfection of HEK293 with mutant construct showed increased size of PG (90 kDa) due to ubiquitin binding (WB); cytoplasmic localization of mutant-PG (IF); higher proliferation and lower apoptosis; fewer and smaller desmosomes in mutant PG cells (EM ) [64] -Additional binding properties of mutant PG to TAIP-2 and HRC-BP (Co-IP and yeast-two-hybrid) [64] - Diminished cell stiffness, but not cell adhesion [65].	Match	n.o.	(het) ACM
c.201del	p.(Ser68Ala fs*92)	Head domain	P	>PTC > NMD, no protein	*Loss of PG?:* Highly reduced levels of *JUP* mRNA (normal splicing), no WB performed [66]	-Absence of PG protein staining in skin biopsies (IF); small desmosomes and wide intercellular spaces (EM) [66].	Match	(hom)EBS; (hom)PPK; (hom) alopecia	(hom) ACM
c.469-8_469 -1del	p.(Val157_Lys161del)	Armadillo Domain 1	LP	>Protein expressed -MutPred-Indel> Pathogenic (0.78939)	*Unclear;* 15 nucleotides shorter cDNA fragments when compared to controls, no WB performed [67].	Cryptic splice acceptor site activation in exon 4 [67].	Unclear	n.r.	(het) ACM
c.1615C > T	p.(Gln539*)	Armadillo Domain 10	P	>PTC > NMD, no protein - MutPred-LOF> Borderline pathogenic (0.62246)	*Loss of PG;* No truncated, or full-length PG protein detected in patient’s skin extracts (WB) [68]. -Apart from the strong reduction of *JUP* (90% reduction), *DSP* and *DSG1* mRNAs were also markedly decreased [68].	-Complete loss of PG protein in the patient’s skin (WB and IF, both with N-terminal and C-terminal antibody); No skin barrier formation; significant reduction of DP and DG3 in patient skin (IF). Only few, abnormal desmosomes were formed [68]. - Strong reduction PG in the myocardium [68].	Match	(hom) lethal EBS	n.o. at young age
c.1729C > T	p.(Arg577Cys)	Armadillo Domain 10	LP	>Protein expressed -PolyPhen-2> Probably damaging (1.000) -SIFT> Tolerated -MutPred2> Pathogenic (0.720)	*Altered PG function:* Mutant PG had similar size as WT and was not reduced on blot according to the study [69].	-On WB, DG2 and Cx43 protein levels were reduced in mutant expression cells, and desmosomal junctions were destabilized (transfection studies) [69].	Match	n.r.	(het) ACM
c.2038_2039del	p.(Trp680Gly fs*11)	Tail domain	P	>PTC > NMD, no protein	*Altered PG function;* C-terminal truncated PG protein is abundantly expressed (56 aa missing) (WB of biopsied LV and RV of multiple patients) [11,65,70].	-Reduced Cx43 and PG in patient ventricles and absence of phosphorylated Cx43 (IF; WB). A decreased number of gap junctions in patient’s myocardium (EM) [70]. - Diminishes cell adhesion, but not the cell compliance [65]. -Mouse *JUP* ^knockin c.2038_2039del^ [71]. -The mouse data contradict the human data and suggest that mutant mRNA is broken down by NMD in mice, and not much protein is produced (WB does show truncated protein). The authors reason that although the deletion is located in exon 11, the PTC is located in the terminal exon (exon 12) > homozygous mice die at postnatal day 1, while cardiac development went normal, mice had severe skin fragility. -Fusion of the last 5 exons in mice, produced the truncated protein fully, did not cause lethality; however, mice did not develop cardiac dysfunction at 11 months of age.	Mismatch Mismatch human: mice	(hom)PPK; (hom)WH [72]	(hom) ACM [72]
c.2057_2058del	p.(Met686Asn fs*5)	Tail domain	n.a.	>PTC > NMD, no protein	*Altered PG function;* Transfected myocytes showed a C-terminal 75 kDa truncated protein (WB) [73].	Cardiac specific Zebrafish *JUP*^WT/2057_2058del^ [73]. - The zebrafish mutated myocytes showed significant reduction of I_Na_ and I_K1_ current densities. EM showed disruption of cell–cell contact. (GAL4/UAS transactivation system was used to induce cardiac specific expression of the human 2057_2058del variant in zebrafish) [73].	Mismatch, but no patients were traced.	n.r.	(hom) ACM claimed, but no patients were traced

Abbreviations: altered protein function > variant annotated in red; partial loss of protein = variant annotated in blue; unclear > variant annotated in grey; US (uncertain significance); LP (likely pathogenic); P (pathogenic); aa (amino acids); n.r. (none reported); n.o. (none observed); NMD (nonsense mediated mRNA decay); fs (frameshift); * or PTC (premature termination codon); WB (Western blot); IF (immunofluorescence); EM (electron microscopy); co-IP (co-immunoprecipitation); WT (wildtype); LV (left ventricle); RV (right ventricle); EBS (epidermolysis bullosa simplex); PPK (palmoplantar keratoderma); WH (woolly hair); ACM (arrhythmogenic cardiomyopathy); hom (homozygous > phenotype observed in homozygous carriers); het (heterozygous > phenotype observed in heterozygous carriers).

**Table 3 ijms-23-10765-t003:** Functionally analysed *DSC2* variants.

HGVS Nomenclature (DNA)	HGVS Nomenclature (Protein)	Protein Domain	ACMG Class	In Silico Predictions	Functional mRNA and Protein Studies	Biological Effect	Prediction: Functional	Skin	Heart
c.-1445G > C	NC_000018.10:g.31103416C > G	5′UTR	B	Not applicable (5′UTR), cannot be predicted	*Altered DC2 function;* n.s. -Luciferase assay >a decreased transcriptional activity for HEK cells transfected with the DC2 mutant (c.-1445C) construct [79].	Altered transcription factor binding in the presence of the mutant allele.	Mismatch by definition	n.r.	(het) ACM
c.140_147del	p.(Lys47Arg fs*2)	PRO peptide-domain	LP	>PTC > NMD, no protein	*Partial loss of DC2:* Patient had reduced levels (>50%) of DC2 in skin biopsy (WB) [80].	n.r. -of note: a relative with only *DSC2*:c.1559T > C (missense) had no phenotype [80].	Match	n.r.	(comp.het) NCCM/HCM with *DSC2*:c.1559T > C
c.304G > A	p.(Glu102Lys)	PRO peptide-domain	LB	>Protein expressed -PolyPhen-2> Benign (0.016) -SIFT> Tolerated -MutPred2> Benign (0.158)	*Altered DC2 function;* n.s., but mutant expressed in cells.	IF shows that the mutant protein localizes in a dotted pattern predominantly in the cytoplasm (COS-1 cells, neonatal rat CM transfection) [81].	Match	n.r.	(het) ACM
c.394C > T	p.(Arg132Cys)	In between PRO peptide and EC1-domain	US	>Protein expressed -PolyPhen-2> Probably damaging (1.000) -SIFT> NOT tolerated -MutPred2> Pathogenic (0.823)	*Partial loss of DC2;* 50% reduced levels of *DSC2* mRNA in explanted heart and hiPSC-CMs; reduced DC2 protein in explanted heart (WB) [82]	-Reduced levels of all desmosomal genes in explanted heart and hiPSC-CMs, reduction of PG at ID in heart; mutant hiPSC-CMs had shortened action potential durations associated with reduced calcium current density and increased potassium current density [82]. -Increased PPARƴ expression and contractile and electric disturbances observed in patient hiPSC-CMs [83] -Zebrafish *DSC2*^WT/c.394C > T^ [82].	Mismatch	n.r.	(het) ACM
c.609C > T	p.(Arg203Cys)	EC1 domain	US	>Protein expressed -PolyPhen-2> Probably damaging (1.000) -SIFT> NOT tolerated -MutPred2> Pathogenic (0.899)	*Altered DC2 function;* Complete defect in processing into the mature form [84]. (WB)	-The mutant protein remains in an unprocessed pro-protein form (COS-1 cells transfection) [84]. -In HL-1 cells, the mutant protein fails to localize at the desmosomes of intercalated disc structures [84].	Match	n.r.	(het) ACM
c.631-2A > G	p.(Ile211Met fs*11)	EC1 domain	P	>PTC > NMD, no protein	*Partial loss of DC2:* Patient heart tissue shows 60% DC2 reduction (WB) [85] -Reduced mRNA *DSC2* (only 3% compared to WT) [85]	-n.r. -Zebrafish KD of *DSC2* and *DSC2*^WT/631−2A > G^ [85].	Match	n.o.	(het) ACM
c.824C > T	p.(Thr275Met)	EC2 domain	US	>Protein expressed -PolyPhen-2> Probably damaging (0.999) -SIFT> NOT tolerated -MutPred2> Benign (0.563)	*Altered DC2 function;* Partial defects in processing into the mature form [84]. (WB)	-Only a proportion of the partly functional DC2 mutant protein is still incorporated into the desmosomes; affects PG at the intercalated disc (COS-1 cells transfection) [84].	Match	n.r.	(hom) ACM
c.1034T > C	p.(Ile345Thr)	EC2 domain	US	>Protein expressed -PolyPhen-2> Possibly damaging (0.756) -SIFT> NOT tolerated -MutPred2> Benign (0.591)	*Altered DC2 function;* n.s., but mutant expressed in cells.	-In transfected neonatal rat cardiomyocytes and HL-1 cells, the mutant protein localizes in the cytoplasm (IF) [81].	Match	n.r.	(het) ACM
c.1559T > C	p.(Ile520Thr)	EC4 domain	LB	>Protein expressed -PolyPhen-2> Probably damaging (0.973) -SIFT> NOT tolerated -MutPred2> Benign (0.601)	*Altered DC2 function;* Protein is expressed, similar size as wildtype (WB) [80]. Unsure if the protein is really altered, or that it maintains all functions [80].	n.r. -of note: a relative with only *DSC2*:c.1559T > C (missense) had no phenotype [80].	Match	n.r.	(comp.het) NCCM/HCM with *DSC2*:c.140_147del
c.1660C > T	p.(Gln554*)	EC4 domain	P	>PTC > NMD, no protein -MutPredLOF> borderline pathogenic (0.51161)	*Altered DC2 function;* Truncated DC2 protein (75 kDa), wildtype is 150 kDa (transfection WB). Less mature protein, more pre-protein than normal [86].	-Heart biopsies shows DC2 staining in hom-carriers (protein is expressed); mutant protein localizes only partially at cell membrane and predominantly in cytoplasm (transfection IF WB) [86]. -Transfected cells show that the secreted truncated isoforms are not anchored in the plasma membrane [87].	Mismatch	Mild PPK at pressure points, in one hom- and one het-carrier (possibly secondary to farm work)	(hom) ACM (LV affected mainly)
c.1841del	p.(Ser614Ile fs*11)	EA domain	P	>PTC > NMD, no protein	*Unclear?;* Truncated isoforms expressed (transfection IF, WB), but needs patient cell confirmation.	Transfected cells show that the secreted truncated isoforms are not anchored in the plasma membrane [87].	Unclear	(hom) Mild PPK, WH [88]	(hom) ACM [88]
c.1913_1916 del	p.(Gln638Leu fs*9)	EA domain	P	>PTC > NMD, no protein	*Partial loss of DC2:* Strong DC2 protein reduction in patient heart tissue (<10% left-WB, also IF) [87]. -*DSC2* mRNA was decreased in patient heart tissue (qPCR)	-Patients’ explanted heart shows degradation of sarcomeres and mitochondria; widened intercellular spaces and accumulation of lipid droplets (EM); Transfected cells show that the secreted truncated isoforms are not anchored in the plasma membrane [87].	Match	n.o.	(hom) ACM
c.2368_2370 del	p.(Gly790del)	In between IA and ICS domains	US	>Protein expressed -MutPred-Indel> NOT pathogenic (0.4309)	*Altered DC2 function:* No reduction of DC2 protein levels [89].	-Slight LV dysfunction with abnormal calcium release [89]. -Mouse model > [89] Hom-mice (G790del) showed enlarged LV and a decreased fractional shortening. Abnormal intracellular calcium release, but no clear ACM phenotype. Het-mice showed no arrhythmias.	Match	n.r.	(het) CM
c.2553del	p.(Asp852Thr fs*4)	ICS domain DC2a only	US	-PTC > terminal exon, not NMD > protein expressed	*Altered DC2a function;* Truncation of the last 47 aa of the DC2a isoform [90].	The mutant protein DC2a lost its ability to bind to PG (HL-1 cells transfection) [90].	Match	n.r.	(het) ACM
c.2687_2688 insGA	p.(Ala897Lys fs*4)	ICS domain DC2a only	B	-PTC > terminal exon, not NMD > protein expressed	*Altered DC2a function;* -Premature termination of the protein [91]. -Does not exhibit defects in processing into the mature form [84].	-Cytoplasmic localization of the mutant protein (HL-1 cells transfection) [91]. -This mutant protein is processed into its mature form and can be incorporated into desmosomes; impaired binding to DP and PG (COS-1 cells transfection) [84]/	Match	n.r.	(het) ACM

Abbreviations: altered protein function > variant annotated in red; partial loss of protein = variant annotated in blue; unclear > variant annotated in grey; US (uncertain significance); B (benign); LB (likely benign); LP (likely pathogenic); P (pathogenic); n.s. (not specified); aa (amino acids); n.r. (none reported); n.o. (none observed); WT (wildtype); WB (Western blot); IF (immunofluorescence); EM (electron microscopy); NMD (nonsense mediated mRNA decay); fs (frameshift); * or PTC (premature termination codon); hiPSC-CMs (human induced pluripotent stem cell derived cardiomyocytes); KD (knockdown); ID (intercalated disc); PPK (palmoplantar keratoderma); CM (cardiomyopathy); ACM (arrhythmogenic cardiomyopathy); NCCM (non-compaction cardiomyopathy); HCM (hypertrophic cardiomyopathy); LV (left ventricle); hom (homozygous > phenotype observed in homozygous carriers); comp.het (compound heterozygous > phenotype observed in compound heterozygous carriers); het (heterozygous > phenotype observed in heterozygous carriers).

**Table 4 ijms-23-10765-t004:** Functionally analysed *KLHL24* variants.

HGVS Nomenclature (DNA)	HGVS Nomenclature (Protein)	Protein Domain	ACMG Class	In Silico Predictions	Functional mRNA and Protein Studies	Biological Effect	Prediction: Functional	Skin	Heart
c.1A > T	p.(Val2_Met29del)	Pre-N-terminal	P	>No protein, loss of start-codon -PolyPhen-2> Benign (0.267) -SIFT> NOT tolerated -MutPred2> Pathogenic (0.881)	*Altered KLHL24 function;* N-terminally truncated protein (28 aa shorter protein, transfection WB). Translation initiation at Met29.	More stable KLHL24 mutant protein which no longer undergoes autoubiquitination. Excessive ubiquitination keratin 14 [98,100]	Mismatch	(het) EBS; (het) PPK	(het) DCM
c.1A > G	p.Val2_Met29del	Pre-N-terminal	P	>No protein, loss of start-codon -PolyPhen-2> Benign (0.267) -SIFT> NOT tolerated -MutPred2> Pathogenic (0.898)	*Altered KLHL24 function;* N-terminally truncated protein (28 aa shorter protein, transfection WB). Translation initiation at Met29. mRNA expression levels not different from WT [98,100,101]	-More stable KLHL24 mutant protein which no longer undergoes autoubiquitination. Excessive ubiquitination keratin 14 [98,100]. -Excessive breakdown of keratin 14 in patient foetal-like hiPSC-derived keratinocytes, otherwise not observed in adult primary keratinocytes [102]. -Excessive ubiquitination of desmin (10% desmin levels left in hiPSC-derived dynamic EHTs). Both RNAi of KLHL24 and overexpression of desmin rescued the in vitro phenotype [101].	Mismatch	(het) EBS; (het) PPK; (het) Alopecia	(het) DCM
c.2T > C	p.Val2_Met29del	Pre-N-terminal	P	> No protein, loss of start-codon -PolyPhen-2> Possibly damaging (0.599) -SIFT> NOT tolerated -MutPred2> Pathogenic (0.929)	*Altered KLHL24 function;* N-terminally truncated (28 aa shorter protein, transfection WB); Translation initiation at Met29. mRNA expression levels not different from WT.	-More stable KLHL24 mutant protein which no longer undergoes autoubiquitination. Excessive ubiquitination keratin 14 [98,100] -Keratins 7,8,17 and 18 were also subjected to KLHL24-mediated degradation in a foetal keratinocyte cell model [103].	Mismatch	(het) EBS; (het) PPK; (het) Alopecia	(het) DCM
c.3G > A	p.Val2_Met29del	Pre-N-terminal	P	> No protein, loss of start-codon -PolyPhen-2> Benign (0.380) -SIFT> NOT tolerated -MutPred2> Pathogenic (0.919)	*Altered KLHL24 function;* N-terminally truncated protein (28 aa shorter protein, transfection WB). Translation initiation at Met29.	More stable KLHL24 mutant protein which no longer undergoes autoubiquitination. Excessive ubiquitination keratin 14 [98,100]	Mismatch	(het) EBS; (het) PPK	(het) DCM
c.3G > T	p.Val2_Met29del	Pre-N-terminal	P	> No protein, loss of start-codon -PolyPhen-2> Benign (0.380) -SIFT> NOT tolerated -MutPred2> Pathogenic (0.919)	*Altered KLHL24 function;* N-terminally truncated protein (28 aa shorter protein, transfection WB). Translation initiation at Met29.	-More stable KLHL24 mutant protein which no longer undergoes autoubiquitination. Excessive ubiquitination keratin 14 [98,100] Excessive ubiquitination of keratin 15 in hair follicle stem cells causes alopecia in mice [104]. Mouse *KLHL24*^WT/3G > T^ [98,104]	Mismatch	(het) EBS; (het) PPK; (het) Alopecia	(het) DCM
c.22A > T	p.Val2_Met29del not p.(Arg8*)	N- terminal	P	>PTC > NMD, no protein	*Altered KLHL24 function;* N-terminally truncated protein (28 aa shorter protein, transfection WB). Translation re-initiation at Met29 after uORF (Met1_Arg8*) [105]	More stable KLHL24 mutant protein. Excessive ubiquitination of desmin (IF) [105].	Mismatch	(het) EBS	(het) DCM
c.917G > A	p.(Arg306His)	before KR	US	>Protein expressed -PolyPhen-2> Probably damaging (0.999) -SIFT> NOT tolerated -MutPred2> Pathogenic (0.740)	n.r. (*loss of KLHL24 suggested*) It is unclear if this protein is expressed or degraded (no WB), but protein activity is expected to be lost.	Higher desmin expression in myocardial biopsies (WB). Ubiquitination loss Desmin? [106,107] (KD *KLHL24* zebrafish) [106]	Unclear	n.o.	(hom) HCM
c.1048G > T	p.(Glu350*)	Kelch repeat (KR1)	P	>PTC > NMD, no protein -MutPred-LOF> NOT pathogenic (0.30843)	n.r. *(loss of KLHL24 suggested)* It is unclear if this protein is expressed or degraded (no WB), but protein activity is lost.	Higher desmin expression in myocardial biopsies (WB). Ubiquitination loss Desmin? [106,107] (KD *KLHL24* zebrafish) [106]	Probable match	n.o.	(hom) HCM

Abbreviations: altered protein function > variant annotated in red; partial loss of protein = variant annotated in blue; US (uncertain significance); P (pathogenic); aa (amino acids); n.r. (none reported); n.o. (none observed); WT (wildtype); WB (western blot); NMD (nonsense mediated mRNA decay); * or PTC (premature termination codon); hiPSC (human induced pluripotent stem cells); EHTs (engineered heart tissues); RNAi (RNA interference); KD (knock down); uORF (upstream open reading frame); EBS (epidermolysis bullosa simplex); PPK (palmoplantar keratoderma); alopecia (hair loss); DCM (dilated cardiomyopathy); HCM (hypertrophic cardiomyopathy); hom (homozygous > phenotype observed in homozygous carriers); het (heterozygous > phenotype observed in heterozygous carriers).

**Table 5 ijms-23-10765-t005:** Functionally analysed *GJA1* variants.

HGVS Nomenclature (DNA)	HGVS Nomenclature (Protein)	Protein Domain	ACMG Class	In Silico Predictions	Functional mRNA and Protein Studies	Biological Effect	Prediction: Functional	Skin (PPK and Hair Only)	Heart
c.5G > T	p.(Gly2Val)	N-terminal cytoplasmic domain	US	>Protein expressed -PolyPhen-2> Probably damaging (0.992) -SIFT> NOT tolerated -MutPred2> Pathogenic (0.769)	*Altered Cx43 function;* n.r., but mutant is expressed.	Impaired channel transfer function. Mutant protein inhibits wt-Cx43.Unable to form functional channels [118].	Match	n.r.	n.r.
c.7G > A	p.(Asp3Asn)	N-terminal cytoplasmic domain	US	>Protein expressed -PolyPhen-2> Probably damaging (0.993) -SIFT> NOT tolerated -MutPred2> Benign (0.413)	*Altered Cx43 function;* n.r., but mutant is expressed.	-Impaired channel transfer function [118,119]. -Mutant protein inhibits wt-Cx43 [118]. -Deficient fibroblast migration, proliferation and differentiation (impaired healing) [119]	Match	n.r.	n.r.
c.19T > G	p.(Leu7Val)	N-terminal cytoplasmic domain	US	>Protein expressed -PolyPhen-2> Probably damaging (0.999) -SIFT> NOT tolerated -MutPred2> Pathogenic (0.662)	*Altered Cx43 function;* n.r., but mutant is expressed. -Greatly reduced total Cx43 levels (<25% of WT) (WB) -Significantly reduced *GJA1* mRNA expression (<25% of WT) (qPCR)	-Impaired phosphorylation. Shortened gap junction length. Severely reduced intercellular electrical coupling. Dysregulation of cell growth, migration and polarization [120]. -Impaired channel transfer function. Mutant protein inhibits wt-Cx43 [118].	Match	n.r.	n.r.
c.23G > T	p.(Gly8Val)	N-terminal cytoplasmic domain	LP	>Protein expressed -PolyPhen-2> Probably damaging (0.913) -SIFT> Tolerated -MutPred2> Benign (0.566)	*Altered Cx43 function;* n.r., but mutant is expressed.	-Augmented hemichannel activity [121]. -Increased keratinocyte cell death and apoptosis [122].	Match	(het) PPK; (het) alopecia	n.r.
c.32T > C	p.(Leu11Pro)	N-terminal cytoplasmic domain	US	>Protein expressed -PolyPhen-2> Probably damaging (1.000) -SIFT> NOT tolerated -MutPred2> Pathogenic (0.951)	*Altered Cx43 function;* n.r., but mutant is expressed.	Translocation of protein from the junctions to the cytoplasm and Golgi network. Impaired channel transfer function. Mutant protein inhibits wt-Cx43 [118].	Match	(het) WH; abnormal hair shaft; keratoderma	n.r.
c.50A > C	p.(Tyr17Ser)	N-terminal cytoplasmic domain	US	>Protein expressed -PolyPhen-2> Probably damaging (0.955) -SIFT> NOT tolerated -MutPred2> Pathogenic (0.889)	*Altered Cx43 function;* n.r., but mutant is expressed.	-Impaired channel transfer function [118,123]. -Complete loss of intercellular electrical coupling [123]. -Inhibited migration [124]. -Translocation of protein from the junctions to the cytoplasm and Golgi network. Mutant protein inhibits wt-Cx43 [118].	Match	n.r.	n.r.
c.52T > C	p.(Ser18Pro)	N-terminal cytoplasmic domain	LP	>Protein expressed -PolyPhen-2> Probably damaging (1.000) -SIFT> NOT tolerated -MutPred2> Pathogenic (0.946)	*Altered Cx43 function;* n.r., but mutant is expressed.	Translocation of protein from the junctions to the cytoplasm and Golgi network. Impaired channel transfer function. Mutant protein inhibits wt-Cx43 [118].	Match	n.r.	n.r.
c.61G > A	p.(Gly21Arg)	TM1	LP	>Protein expressed -PolyPhen-2> Probably damaging (0.972) -SIFT> NOT tolerated -MutPred2> Pathogenic (0.784)	*Altered Cx43 function;* n.r., but mutant is expressed.	-Complete loss of intercellular electrical coupling [123]. -Reduced channel conductance [125,126]. -Mutant protein inhibits wt-Cx43 [125,126,127]	Match	n.r.	(het) Heart developmental formation
c.93T > G	p.(Ile31Met)	TM1	US	>Protein expressed -PolyPhen-2> Probably damaging (0.980) -SIFT> NOT tolerated -MutPred2> Pathogenic (0.854)	*Altered Cx43 function;* n.r., but mutant is expressed.	Impaired phosphorylation. Inhibited gap junctional coupling. Augmented hemichannel activity [128].	Match	n.r.	n.r.
c.97C > T	p.(Arg33*)	TM1	LP	-PTC- only 1 exon, not NMD > protein expressed -MutPred-LOF> Pathogenic (0.64454)	*Altered Cx43 function;* n.r., but mutant is expressed.	-Translocation of protein from the junctions to the cytoplasm and nucleus. Unable to form functional channels [129].	Match	n.r.	n.r.
c.119C > T	p.(Ala40Val)	TM1	US	>Protein expressed -PolyPhen-2> Probably damaging (0.990) -SIFT> NOT tolerated -MutPred2> Pathogenic (0.841)	*Altered Cx43 function;* n.r., but mutant is expressed.	Impaired channel transfer function. Complete loss of intercellular electrical coupling [123].	Match	n.r.	n.r.
c.124G > A	p.(Glu42Lys)	E1	LP	>Protein expressed -PolyPhen-2> Probably damaging (0.993) -SIFT> NOT tolerated -MutPred2> Pathogenic (0.885)	*Altered Cx43 function;* n.r., but mutant is expressed.	-Complete loss of intercellular electrical coupling. Areas with complete loss of protein signal in immunohistochemical staining (mosaicism) [130] *-GJA1*^c.124G > A/WT^ mouse model [131]	Match	n.r.	(het) Cardiac arrhythmias
c.131C > T	p.(Ala44Val)	E1	LP	>Protein expressed -PolyPhen-2> Benign (0.140) -SIFT> Tolerated -MutPred2> Pathogenic (0.709)	*Altered Cx43 function;* n.r., but mutant is expressed.	-Augmented hemichannel activity [121]. -Translocation of protein from the keratinocyte junctions to the cytoplasm [13].	Match	(het) PPK; skin keratosis, erythema and naevus	n.r.
c.145C > A	p.(Gln49Lys)	E1	LP	>Protein expressed -PolyPhen-2> Probably damaging (0.955) -SIFT> NOT tolerated -MutPred2> Pathogenic (0.866)	*Altered Cx43 function;* n.r., but mutant is expressed.	Impaired channel transfer function. Mutant protein inhibits wt-Cx43 [126].	Match	n.r.	n.r.
c.154_156 dup	p.(Phe52dup)	E1	US	>Protein expressed -MutPred-Indel> Pathogenic (0.80091)	*Altered Cx43 function;* n.r., but mutant is expressed.	Unable to produce functional gap junction plaques [123].	Match	n.r.	n.r.
n.r. no patients reported with this variant.	p.(Gly60Ser)	E1	n.a.	>Protein expressed -PolyPhen-2> Probably damaging (0.999) -SIFT> NOT tolerated -MutPred2> Pathogenic (0.915)	*Altered Cx43 function;* Reduced protein levels of both phosphorylated and total Cx43 protein (WB) [132,133,134,135,136,137].	-Muscle gap junction intracellular communication severely reduced [132]. -Impaired phosphorylation [138]. -Reduced number of gap junction plaques [133,135]. -Translocation of protein from the junctions to the cytoplasm and Golgi network [133,135,136]. - Reduced intercellular electrical coupling [133,135,137]. -Mutant protein reduces phosphorylation of wt-Cx43 [133,134]. -Arrythmias [139]. -Defective smooth muscle contraction. Impaired reprogramming in response to stretch in smooth muscle [134]. -Impaired bone and teeth development [135,140]. -Increased infarct damage after cerebral artery occlusion [136]. -Enhanced keratinocyte proliferation and differentiation [141]. -Deficient fibroblast migration, proliferation and differentiation (impaired wound healing) [119] -Disturbed hair growth [142]. *-GJA1*^G60S/WT^ mouse models [134,136,137,138,141,142,143,144,145,146]	Match	WH/ Alopecia in mice.	n.r.
c.205T > C	p.(Ser69Pro)	E1	US	>Protein expressed -PolyPhen-2> Possibly damaging (0.869) -SIFT> NOT tolerated -MutPred2> Pathogenic (0.911)	*Altered Cx43 function;* n.r., but mutant is expressed.	Total loss of membrane transfer capability [147].	Match	n.r.	n.r.
c.227G > A	p.(Arg76His)	TM2	US	>Protein expressed -PolyPhen-2> Possibly damaging (0.690) -SIFT> NOT tolerated -MutPred2> Pathogenic (0.743)	*Altered Cx43 function;* n.r., but mutant is expressed.	Reduced channel conductance [129].	Match	(het) Alopecia	n.r.
c.268C > G	p.(Leu90Val)	TM2	US	>Protein expressed -PolyPhen-2> Possibly damaging (0.817) -SIFT> NOT tolerated -MutPred2> Pathogenic (0.695)	*Altered Cx43 function;* n.r., but mutant is expressed.	-Impaired channel transfer function [123]. -Reduced channel conductance [123,126]. -Mutant protein inhibits wt-Cx43 [126].	Match	n.r.	n.r.
c.389T > C	p.(Ile130Thr)	Cytoplasmic loop domain	LP	>Protein expressed -PolyPhen-2> Benign (0.013) -SIFT> Tolerated -MutPred2> Benign (0.506)	*Altered Cx43 function*: Reduced Cx43 protein total and phosphorylated levels compared to WT (WB) [134,138,148].	-Impaired channel transfer function [123,134]. -Reduced channel conductance [123,148]. -Defective smooth muscle contraction. Impaired reprogramming in response to stretch in smooth muscle [134]. -Impaired phosphorylation [138]. -Slower cardiac conduction velocity. Electrocardiogram abnormalities and arrhythmias in vivo and ex vivo [148] *-GJA1*^I130T/WT^ mouse model [134,138,144,148]	Match	n.r.	(het) Ventricular tachycardia; Arrhythmia Atrioventricular block [149]
c.400A > G	p.(Lys134Glu)	Cytoplasmic loop domain	US	>Protein expressed -PolyPhen-2> Benign (0.002) -SIFT> Tolerated -MutPred2> Benign (0.570)	*Altered Cx43 function;* n.r., but mutant is expressed.	Impaired channel transfer function. Reduced channel conductance [123].	Match	(het) PPK	n.r.
c.412G > C	p.(Gly138Arg)	Cytoplasmic loop domain	US	>Protein expressed -PolyPhen-2> Probably damaging (0.985) -SIFT> Tolerated -MutPred2> Benign (0.462)	*Altered Cx43 function;* n.r., but mutant is expressed. -Significantly reduced total Cx43 (<70% of WT) and phosphorylated levels (WB) [120,150].	-Impaired phosphorylation [120,128]. -Shortened gap junction length [128]. -Reduced number of gap junction plaques [148]. -Reduced channel conductance [125]. -Dysregulation of cell growth, migration and polarization [120]. -Mutant protein inhibits wt-Cx43 [125,127]. -Unable to form functional channels [127,150]. -Electrocardiogram abnormalities and arrhythmias in vivo and ex vivo [150]. -Inhibited gap junctional coupling [128,148]. -Augmented hemichannel activity. Increased half-time of Cx43 protein [128]. -Altered expression of Shh and Bmp2 [151]. *GJA1*^WT/c.412G > C^ mouse model [150,151].	Match	(het) PPK; (het) Alopecia	(het) Arrhythmia.
c.427G > A	p.(Gly143Ser)	Cytoplasmic loop domain	US	>Protein expressed PolyPhen-2> Possibly damaging (0.788) -SIFT> Tolerated -MutPred2> Benign (0.604)	*Altered Cx43 function;* n.o. mutant is expressed, but size and amount is similar at WT (WB). -mRNA levels are similar as WT.	-Impaired phosphorylation [120,128]. -Dysregulation of cell growth, migration and polarization [120]. -Inhibited gap junctional coupling. Augmented hemichannel activity. Increased half-time of Cx43 protein [128].	Match	(het) WH/ Alopecia	n.r.
c.440T > C	p.(Met147Thr)	Cytoplasmic loop domain	US	>Protein expressed PolyPhen-2> Probably damaging (0.979) -SIFT> NOT tolerated -MutPred2> Pathogenic (0.822)	*Altered Cx43 function;* n.r., but mutant is expressed (transfection).	-Retains α-helical structure, but exhibits loss of function and cellular trafficking defects [152].	Match	(het) Alopecia	n.r.
c.443G > A	p.(Arg148Gln)	Cytoplasmic loop domain	US	>Protein expressed PolyPhen-2> Possibly damaging (0.773) -SIFT> Tolerated -MutPred2> Benign (0.442)	*Altered Cx43 function;* n.r., but mutant is expressed (transfection).	-Retains α-helical structure, but exhibits loss of function and cellular trafficking defects [152].	Match	n.r.	n.o.
c.460A > G	p.(Thr154Ala)	Cytoplasmic loop domain	LP	>Protein expressed PolyPhen-2> Possibly damaging (0.662) -SIFT> NOT tolerated -MutPred2> Pathogenic (0.660)	*Altered Cx43 function;* n.r. but mutant is expressed (transfection).	-Retains α-helical structure, but exhibits loss of function and cellular trafficking defects [152].	Match	(het) dry hair, thin skin	(het) Ventricular septal defects; extrasystoles
c.581A > C	p.(His194Pro)	E2	LP	>Protein expressed -PolyPhen-2> Probably damaging (0.939) -SIFT> NOT tolerated -MutPred2> Pathogenic (0.909)	*Altered Cx43 function;* n.r., but mutant is expressed.	Impaired phosphorylation. Inhibited gap junctional coupling. Augmented hemichannel activity [128].	Match	n.r.	n.r.
c.605G > A	p.(Arg202His)	E2	US	>Protein expressed -PolyPhen-2> Probably damaging (1.000) -SIFT> NOT tolerated -MutPred2> Pathogenic (0.819)	*Altered Cx43 function;* n.r., but mutant is expressed.	-Unable to produce functional gap junction plaques. Decreased permeability [123]. -Reduced channel conductance. Translocation of protein from the junctions to the cytoplasm and Golgi network. Mutant protein inhibits wt-Cx43 [126].	Match	n.r.	n.r.
c.646G > T	p.(Val216Leu)	TM4	US	>Protein expressed -PolyPhen-2> Possibly damaging (0.869) -SIFT> NOT tolerated -MutPred2> Pathogenic (0.760)	*Altered Cx43 function;* n.r., but mutant is expressed. -Reduced protein levels in hiPSC (WB) [153]. -Reduced mRNA levels in hiPSC [153].	-Impaired channel transfer function [119,153]. -Deficient fibroblast migration, proliferation and differentiation (impaired wound healing) [119] -Delayed osteoblast differentiation [153]. -Reduced channel conductance. Translocation of protein from the junctions to the cytoplasm and Golgi network. Mutant protein inhibits wt-Cx43 [126].	Match	n.r.	n.r.
c.681A > T	p.(Glu227Asp)	TM4	LP	>Protein expressed -PolyPhen-2> Possibly damaging (0.900) -SIFT> NOT tolerated -MutPred2> Pathogenic (0.811)	*Altered Cx43 function;* n.r., but mutant is expressed.	-Augmented hemichannel activity [121]. -Translocation of protein from the keratinocyte junctions to the cytoplasm [13].	Match	(het) Skin keratosis and erythema	n.r.
n.r.	p.(Lys258*)	C-terminal cytoplasmic domain	n.a.	-PTC > NOT NMD -MutPred-LOF> Pathogenic (0.63488)	*Altered Cx43 function;* n.r., but mutant is expressed.	-Increased keratinization and unstable skin structure [154]. *-GJA1*^K258S*/WT^ mouse model [154,155]	Match	n.r.	n.r.
c.780_781del	p.(Cys260Trp fs*47)	C-terminal cytoplasmic domain	LP	-PTC- only 1 exon, not NMD > protein expressed	*Altered Cx43 function;* Produced a shortened 305 residues instead of a 382 wt-Cx43	Reduced number of gap junction plaques. Severely reduced intercellular electrical coupling. Mutant protein inhibits wt-Cx43 in a dose-dependent manner [156].	Match	(het) PPK	n.r.
c.848C > T	p.(Pro283Leu)	C-terminal cytoplasmic domain	US	>Protein expressed -PolyPhen-2> Probably damaging (0.997) -SIFT> Tolerated -MutPred2> Pathogenic (0.690)	*Altered Cx43 function;* n.r., but mutant is expressed.	-Translocation of protein from the keratinocyte junctions to the cytomembrane and cytoplasm [157]. - Formed functional gap junction channels, with no evidence of augmented hemichannel function or induction of cell death (transfection) [158].	Match	(het) PPK; skin keratosis and erythema	n.r.
c.869C > A	p.(Thr290Asn)	C-terminal cytoplasmic domain	US	>Protein expressed -PolyPhen-2> Probably damaging (0.976) -SIFT> Tolerated -MutPred2> Benign (0.308)	*Altered Cx43 function;* n.r., but mutant is expressed.	-Translocation of protein from the intracellular keratinocyte junctions to the cytomembrane and cytoplasm [157]. -Formed functional gap junction channels, with no evidence of augmented hemichannel function or induction of cell death (transfection) [158].	Match	(het) PPK; skin keratosis and erythema	n.r.
c.932del	p.(Ala311Val fs*37)	C-terminal cytoplasmic domain	US	-PTC- only 1 exon, not NMD > protein expressed	*Altered Cx43 function;* n.r., but mutant is expressed.	-Significant inhibition of membrane transfer capability [147]. -Translocation of protein from the junctions to the cytoplasm and Golgi network. Severely reduced intercellular electrical coupling. Genetic mosaicism [159].	Match	n.r.	(het) Atrial fibrillation [159]
c.977C > T	p.(Thr326Ile)	C-terminal cytoplasmic domain	US	>Protein expressed -PolyPhen-2> Probably damaging (0.975) -SIFT> NOT tolerated -MutPred2> Benign (0.326)	*Altered Cx43 function;* n.r., but mutant is expressed.	Total loss of membrane transfer capability [147].	Match	n.r.	n.r.
n.r.	p.(Ser364Pro)	C-terminal cytoplasmic domain	US	>Protein expressed -PolyPhen-2> Benign (0.000) -SIFT> Tolerated -MutPred2> Benign (0.315)	*Altered Cx43 function;* n.r., but mutant is expressed.	-Abnormalities in phosphorylation and cell–cell communication [160]. -Impaired pH sensitivity [161].	Match	n.r.	(het) Heart developmental formation

Abbreviations: altered protein function > variant annotated in red; US (uncertain significance); LP (likely pathogenic); n.r. (none reported); n.o. (none observed); WB (Western blot); fs (frameshift); * or PTC (premature termination codon); NMD (nonsense mediated mRNA decay); PPK (palmoplantar keratoderma); WH (woolly hair); WT (wildtype); E1-E2 (extracellular loop); TM1-M4 (transmembrane segment). Over 90% of variants reported here also give ODDD (oculodentodigital dysplasia) and HL (hearing loss); het (heterozygous > phenotype observed in heterozygous carriers).

## Data Availability

Not applicable.

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
