# Peer review of "Towards a Better Understanding of Genotype–Phenotype Correlations and Therapeutic Targets for Cardiocutaneous Genes: The Importance of Functional Studies above Prediction"

_ijms, 2022, doi:10.3390/ijms231810765_

Round 1
Reviewer 1 Report
I will review here the manuscript 'Towards a better understanding of genotype-phenotype correlations and therapeutic targets for cardiocutaneous genes: the importance of functional studies above prediction.
First I want to congratulate the authors to this impressive review article. The authors developed in this review article a perfect overview about cardiocutaneous diseases and this review article must be published!!! However there are some points, which can still improve the (high) quality of this review article.
1.) Please mention in the abstract the major conclusion of your review article.
2.) Line 44/45. Is it possible to indicate prevalences for the different cardiomyopathies?
3.) Line 50: Please mention that DSP encodes desmoplakin.
4.) Line 52: Could you please add citations for JUP and GJA1?
5.) Does it makes sense to introduce the reader to the terms Naxos disease and/or Carvajal syndrome?
6.) Figure 1 - Legend: You mean proteins not genes!
7.) Could you please explain in more detail how cardiac and skin desmosomes differ in their molecular composition of the different genes? A Figure comparing both kind of desmosomes would be helpful.
8.) Could you add dates for your ClinVar analysis? I think this makes sense, because in some years the data in ClinVar could be extended.
9.) Line 164: Could you specifiy the word altered? What do you exactly mean?
10.) Could you explain the alternative splicing of DSP and DSC2 also at the exon mRNA level not only at the protein level?
11.) Could you please add also the DSC2 mutation found by Simpson et al. (
- PMID: 18957847, DOI: 10.1159/000165696) into Figure 4. I think this mutationis missing. 12. ) You indicate for Q554X a cardiocutaneous phenotype in Figure 4. This is not correct. Please read in detail the paper of Gerull B et al. where they found that the patients are farmers and that the skin phenotype was therefor also present in non-mutation carriers! 13.) Table 3. You should add the secretion of the truncated DSC2 which was described by Brodehl et al. for three different truncating DSC2 variants. 14.) Please add the phenotype of the mouse model for p.G790del in Table 3. 15.) Please extented the abnormal secretion of truncated DSC2 investigated by Brodehl et al. 2020 (
- 10.1016/j.yjmcc.2020.03.006)
- 16.) Line 367: Please add in addition a review about DES mutations causing cardiomyopathies (e.g. Brodehl et al. 2018 (PMID 29926427). Then the interested reader can also follow this gene, also it is not causing a skin phenotype.
- 17.) Does it makes sense to check in addition also the HGMD database for additional variants / mutations?
- In summary, I have enjoyed it to review this nice and super interesting and relevant review article. I hope that I could contribute a little to improve the quality a little bit. Good luck with the revision!
Author Response
Reviewer 1: First I want to congratulate the authors to this impressive review article. The authors developed in this review article a perfect overview about cardiocutaneous diseases and this review article must be published!!! However there are some points, which can still improve the (high) quality of this review article.
- Response: We thank this reviewer for the time to review our manuscript and are delighted to hear it was received with such enthusiasm.
1.) Please mention in the abstract the major conclusion of your review article.
- Response: Unfortunately, we are limited to 200 words for the abstract and have no additional space to elaborate more on the major conclusions of the investigated genes.
2.) Line 44/45. Is it possible to indicate prevalences for the different cardiomyopathies?
- Response: We have changed this sentence to “While arrhythmogenic cardiomyopathy (ACM) is the most commonly observed form, with a prevalence of 1:5000, other cardiomyopathies such as dilated (DCM; 1:250), hypertrophic (HCM; 1:500) or non-compaction (NCCM) have also been observed in patients.” Currently, the prevalence of NCCM is still difficult to estimate and we therefore left this out.
3.) Line 50: Please mention that DSP encodes desmoplakin.
- Response: We have changed this sentence to “In 2000, the first desmosomal gene variant underlying a cardiocutaneous syndrome was identified as a homozygous DSP variant that translated to a C-terminal truncated desmoplakin protein.”
4.) Line 52: Could you please add citations for JUP and GJA1?
- Response: We included several more citations for the genes in question, including for comment 5, below.
5.) Does it makes sense to introduce the reader to the terms Naxos disease and/or Carvajal syndrome?
- Response: We agree that this indeed makes sense to mention these terms (for the well-informed readers) and therefore have added these to the following sentence “Today, many variants have been associated with disease, found in desmosomal genes DSP (also known as Carvajal syndrome), JUP (also known as Naxos Disease) and DSC2, the intermediate filament regulating gene KLHL24, and the gap junction gene GJA1.”
6.) Figure 1 - Legend: You mean proteins not genes!
- Response: This is indeed true and have changed this into the title of the figure. Note that we have denoted both the gene names and corresponding proteins in the legend to provide additional clarity.
7.) Could you please explain in more detail how cardiac and skin desmosomes differ in their molecular composition of the different genes? A Figure comparing both kind of desmosomes would be helpful.
- Response: We understand the reviewers concern. As can be observed in Figure 1, different types (colors) of desmosomes are depicted in the skin. However, as the skin may contain any kind of combination: DPI-II, PG, DC1-3, DG1-4 and PP1-3, this will be more difficult to draw out into a figure. Nonetheless, we have adjusted the text section (“2.Desmosomal genes”) accordingly, to make it more clear.
8.) Could you add dates for your ClinVar analysis? I think this makes sense, because in some years the data in ClinVar could be extended.
- Response: This is indeed a good point, as new variants are indeed continuously added to the ClinVar database. We have added the dates of our analysis to the Material and Method section.
9.) Line 164: Could you specifiy the word altered? What do you exactly mean?
- Response: With the word altered, we mean that the amino acid composition is altered compared to the WT protein. This may or may not have an effect on protein function.
10.) Could you explain the alternative splicing of DSP and DSC2 also at the exon mRNA level not only at the protein level?
- Response: This is indeed a good suggestion and we have added this information in the text.
11.) Could you please add also the DSC2 mutation found by Simpson et al. (PMID: 18957847, DOI: 10.1159/000165696) into Figure 4. I think this mutation is missing.
- Response: The reviewer is indeed correct. We have included this variant into the manuscript, including the studies that provided the functional data. Of note, the overall data is not clear as to whether variant c.1841delG leads to production of protein (detectable levels) in patient cells.
- ) You indicate for Q554X a cardiocutaneous phenotype in Figure 4. This is not correct. Please read in detail the paper of Gerull B et al. where they found that the patients are farmers and that the skin phenotype was therefor also present in non-mutation carriers!
- Response: Excellent observation and we indeed previously missed this. We have adjusted this into the table / text.
13.) Table 3. You should add the secretion of the truncated DSC2 which was described by Brodehl et al. for three different truncating DSC2 variants.
- Response: We have included this data into the table (for DSC2 variants c.1660C>T, c.1841delG and c.1913_1916delAGAA).
14.) Please add the phenotype of the mouse model for p.G790del in Table 3.
- Response: We have included the results from this mouse model in the table.
15.) Please extented the abnormal secretion of truncated DSC2 investigated by Brodehl et al. 2020 ( 10.1016/j.yjmcc.2020.03.006)
- Response: We have included this data into the table (for DSC2 variants c.1660C>T, c.1841delG and c.1913_1916delAGAA).
16.) Line 367: Please add in addition a review about DES mutations causing cardiomyopathies (e.g. Brodehl et al. 2018 (PMID 29926427). Then the interested reader can also follow this gene, also it is not causing a skin phenotype.
- Response: We have included the reference for this review at line 377.
17.) Does it makes sense to check in addition also the HGMD database for additional variants / mutations? In summary, I have enjoyed it to review this nice and super interesting and relevant review article. I hope that I could contribute a little to improve the quality a little bit. Good luck with the revision!
- Response: Currently, we use the ClinVar database to make a rough assessment of the number of reported variants, but we observed that these numbers increase dramatically every month. Addition of variants from the HGMD database will probably make these numbers even higher. However, the point that the total amount of variants with functional evidence is low, is already strongly emphasized through assessment of the ClinVar database, and we therefore did not feel the need to include other databases as well.
We sincerely thank the reviewer for the comments that helped improve this review.

Reviewer 2 Report
In this review, Vermeer, et al summarized the genetic variants causing skin and cardiac phenotypes. Substantial number of genetic variants in DSP, JUP, DSC2, KLHL24, and GJA1 were reanalyzed by the authors using generally available database and algorithm (ClinVar, PubMed, ACMG guidelines, SIFT, PolyPhen-2, MutPred2, etc.), classified into “protein reducing” or “altered protein”, then one-to-one linked with their functional role. This comprehensive review based on the extensive efforts provide a wide-range information of the genetic variants identified in cardiocutaneous syndrome to both clinical and basic researchers at a glance. Therapeutic implications based on each genetic variant and their function are informative. I have some minor comments.
1. Analytical method is thoroughly described in “6. Materials and Methods” section, but readers recognize them at the end of the review. Please add the description specifying the location of “6. Materials and Methods” before Results.
2. The authors compared the prediction of variant pathogenicity and functional effect based on in silico algorithm with the real functional evidence, then classified each finding into Match, Mismatch, Probably match, or Unclear, etc. Addition of the analytical method how the authors defined each classification would be more helpful for the readers.
3. In figure1, desmoglein-2 and plakophilins were not denoted. Because these molecules seem illustrated in the lower-right picture, addition of these gene name may be more informative for the readers not familiar with this research field.
4. Only section headings 3 and 4 consist of capital letters, while the other headings do not.
5. Subheading 4.1.1 requires line feed.
Author Response
Reviewer 2:
In this review, Vermeer, et al summarized the genetic variants causing skin and cardiac phenotypes. Substantial number of genetic variants in DSP, JUP, DSC2, KLHL24, and GJA1 were reanalyzed by the authors using generally available database and algorithm (ClinVar, PubMed, ACMG guidelines, SIFT, PolyPhen-2, MutPred2, etc.), classified into “protein reducing” or “altered protein”, then one-to-one linked with their functional role. This comprehensive review based on the extensive efforts provide a wide-range information of the genetic variants identified in cardiocutaneous syndrome to both clinical and basic researchers at a glance. Therapeutic implications based on each genetic variant and their function are informative. I have some minor comments.
- Response: We thank this reviewer for the time to review our manuscript and are glad to read this reviewer’s positive remarks.
- Analytical method is thoroughly described in “6. Materials and Methods” section, but readers recognize them at the end of the review. Please add the description specifying the location of “6. Materials and Methods” before Results.
- Response: Due to the layout of the IJMS, we do not believe we have influence over the position of the Material and Method section. But we agree with this reviewer that it would make more sense to move this section up. We leave this in the hands of the editorial team of IJMS.
- The authors compared the prediction of variant pathogenicity and functional effect based on in silico algorithm with the real functional evidence, then classified each finding into Match, Mismatch, Probably match, or Unclear, etc. Addition of the analytical method how the authors defined each classification would be more helpful for the readers.
- Response: This is indeed a good point which is currently missing in the material and method section. We have included this classification into this revised manuscript as “Finally, the functional evidence of each variant was compared to the in silico predictions and this was categorized as a ‘match’, if the predictions were on par with the functional evidence observed in patient-derived cells/tissues and or transfection/animal studies; ‘probably match’ if the functional predictions were on par with the functional evidence, but decisive proof of the latter was incomplete; ‘mismatch’ if the functional predictions were not on par with the functional evidence; and ‘unclear’ if the functional evidence itself was inconclusive or contradictory.”
- In figure1, desmoglein-2 and plakophilins were not denoted. Because these molecules seem illustrated in the lower-right picture, addition of these gene name may be more informative for the readers not familiar with this research field.
- Response: We have not denoted the plakophilins and desmogleins into the figure itself because we did not want to confuse the reader by suggesting that variants into these genes also cause a cardiocutaneous syndrome. However, we do see this reviewers valid point and have therefore adjusted the second part of the legend of Figure 1 to “The lower right panel depicts a plasma membrane between two adjacent cells containing a gap junction, connexin 43 (gene GJA1, protein Cx43) and a desmosomal junction, consisting of desmoplakin (gene DSP, protein DP), plakoglobin (gene JUP, protein PG), desmocollins (subtype 2; gene DSC2, protein DC2) desmogleins (not denoted, blue protein structures) and plakophilins (not denoted, red protein structures). Intermediate filaments (not denoted, turquoise protein structures) adhere to the desmosomal junction. Kelch-like protein 24 (gene and protein KLHL24) mostly accumulates near the plasma membrane.”
- Only section headings 3 and 4 consist of capital letters, while the other headings do not.
- Response: We have corrected this in the revised manuscript.
- Subheading 4.1.1 requires line feed.
- Response: We have corrected this in the revised manuscript.
